# A new habitat map of the Lena Delta in Arctic Siberia based on field and remote sensing datasets

Simeon Lisovski[1,*], Alexandra Runge[2,*], Iuliia Shevtsova[1], Nele Landgraf[3], Anne Morgenstern[2], Ronald Reagan Okoth[1,4], Matthias Fuchs[2], Nikolay Lashchinskiy[5,6], Carl Stadie[2,7], Alison Beamish[8], Ulrike Herzschuh[1,9,10], Guido Grosse[1,11], Birgit Heim[1]

[*] Both authors contributed equally

[1] Alfred Wegener Institute Helmholtz Centre for Polar and Marine Research, Polar Terrestrial Environmental Systems, 14473 Potsdam, Germany

[2] Alfred Wegener Institute Helmholtz Centre for Polar and Marine Research, Permafrost Research, 14473 Potsdam, Germany

[3] Humboldt University, Department of Geosciences, 12489 Berlin, Germany

[4] Julius-Maximilians Universität Würzburg, Institute of Geography and Geology, Oswald-Külpe-Weg 86, 97074 Würzburg, Germany

[5] Central Siberian Botanical Garden, Siberian Branch, Russian Academy of Sciences, Novosibirsk, 630090 Russia

[6] Trofimuk Institute of Petroleum Geology and Geophysics, Siberian Branch, Russian Academy of Sciences, Novosibirsk, 630090 Russia

[7] University of Greifswald, Institute for Geography and Geology, Germany (current address: University of Copenhagen, Department of Earth Science and Nature Management, Denmark)

[8] GFZ German Research Centre for Geosciences, Helmholtz Centre Potsdam.

[9] University of Potsdam, Institute of Environmental Sciences & Geography, Karl-Liebknecht-Str. 24-25, 14476 Potsdam, Germany

[10] University of Potsdam, Institute of Biochemistry and Biology, Karl-Liebknecht-Str. 24-25, 14476 Potsdam, Germany

[11] University of Potsdam, Institute of Geosciences, Karl-Liebknecht-Str. 24-25, 14476 Potsdam, Germany

*Correspondence to*: Simeon Lisovski (Simeon.Lisovski@awi.de), Alexandra Runge (alexandra.runge@gfz.de), Birgit Heim (birgit.heim@awi.de)

**Abstract.** The Lena Delta is the largest river delta in the Arctic (about 30 000 km2) and prone to rapid changes due to climate warming, associated cryosphere loss and ecological shifts. The delta is characterized by ice-rich permafrost landscapes and consists of geologically and geomorphologically diverse terraces covered with tundra vegetation and of active floodplains, featuring approximately 6 500 km of channels and over 30 000 lakes. Because of its broad landscape and habitat diversity the delta is a biodiversity hotspot with high numbers of nesting and breeding migratory birds, fish, caribou and other mammals and was designated a State Nature Reserve in 1995. Characterizing plant composition, above ground biomass and application of field spectroscopy was a major focus of a 2018 expedition to the delta. These field data collections were linked to Sentinel-2 satellite data to upscale local patterns in land cover and associated habitats to the entire delta. Here, we describe multiple field datasets collected in the Lena Delta during summer 2018 including foliage projective cover (Shevtsova et al., 2021a), above ground biomass (Shevtsova et al., 2021b), and hyperspectral field measurements (Runge et al., 2022). We further describe a detailed Sentinel-2 satellite image-based classification of habitats for the central Lena Delta (Landgraf et al., 2022), an upscaled classification for the entire Lena Delta (Lisovski et al., 2022), as well as a synthesis product for disturbance regimes (Heim and Lisovski, 2023) in the delta that is based on the classification, the described datasets, and field expertise. We present context and detailed methods of these openly available datasets and show how their combined use can improve our understanding of the rapidly changing Arctic tundra system. The new Lena Delta habitat classification represents a first baseline against which future observations can be compared. The link between such detailed habitat classifications and disturbance regime may provide a better understanding of how Arctic lowland landscapes will respond to climate change and how this will impact land surface processes.

# 1 Introduction

Global warming has profound impacts on the polar regions (Serreze and Barry, 2011; Overland et al., 2019). Rapidly increasing temperatures and changing precipitation regimes result in declining sea ice, warming and thawing of permafrost, more frequent tundra fires, and changes in vegetation (e.g., Biskaborn et al., 2019; Hu et al., 2015; Mauclet et al., 2022; Box et al., 2019; Amap, 2021). The Arctic tundra biome, which is normally characterized by harsh living conditions and nutrient-deficiency, has experienced rapid phenological shifts, such as earlier green-up in spring, which is also associated with increasing shrubification rates (Mekonnen et al., 2021). Shifts in plant communities are also driven by changing nutrient availability in permafrost soils (Mekonnen et al., 2021; Mauclet et al., 2022), affecting the net primary productivity of tundra ecosystems.

Satellite-derived remote sensing can provide large-scale assessments of Arctic vegetation cover and
changes therein (Bartsch et al., 2016). For example, the Circumpolar Arctic Vegetation Map (CAVM)
project, from the Conservation of Arctic Flora and Fauna working group (CAFF), provided a first
panarctic vegetation composition map based on Advanced Very-High Resolution Radiometer
(AVHRR) false-color infrared (CIR) composites at a 1:4 million map scale (Walker, 1998; Raynolds
et al., 2019). Later, higher resolution land cover maps became available across all spatial scales
from national and international efforts such as the NASA Arctic-Boreal Vulnerability Experiment
(ABoVE) providing open-source data collections from boreal and arctic regions (ABoVE Science
Definition Team, 2014) specifically for Alaska, Canada, Northern Europe, and Western Siberia,
providing a better bridge to field measurements. Such products greatly assist in monitoring and
upscaling of patterns and dynamics of soil properties, land-atmosphere fluxes, ecosystem states,
and changes therein (e.g., Walker, 1998; Beamish et al., 2020; Berner et al., 2020; Sweeney et al.,
2022; Macander et al., 2022; Endsley et al., 2022). For selected Eastern Siberian tundra regions,
land cover maps have been produced (e.g., Veremeeva and Gubin, 2009; Bartsch et al., 2019;
Schneider et al., 2009), including the Lena Delta (Bartsch et al., 2019; Schneider et al., 2009).
Arctic river deltas represent distinct and vulnerable geomorphological and ecological regions at the
marine-terrestrial boundary. River deltas have been studied intensively to better understand land
cover and vegetation compositions (Jorgenson, 2000; Schneider et al., 2009; Frost et al., 2020;
Bartsch et al., 2020), carbon pools and fluxes (Bartlett et al., 1992; Schneider et al., 2009; Sachs et
al., 2008; Rossger et al., 2022), and land cover change caused by climate change impacts
(Jorgenson, 2000; Pisaric et al., 2011; Lantz et al., 2015; Nitze and Grosse, 2016; Vulis et al., 2021;
Juhls et al., 2021). With diverse habitats, Arctic river deltas are biodiversity hotspots (Gilg et al.,
2000), but at the same time are prone to rapid changes (Walker, 1998; Overeem et al., 2022). Arctic
deltas are affected by permafrost thaw (e.g., Pisaric et al., 2011; Nitze and Grosse, 2016; Vulis et
al., 2021), sea ice loss (Overeem et al., 2022), and increased sediment transport and organic load
during spring floods (Piliouras and Rowland, 2020; Juhls et al., 2021). Arctic river deltas are very
dynamic systems and high-resolution habitat information from these biodiversity hotspots is needed
to assess and predict changes and implications of Arctic warming.
The Lena Delta is the largest Arctic river delta representing a typical lake-rich lowland permafrost
landscape (Grigoriev, 1993). Over the past decades, the central Lena Delta became a place of
intensive international research. In addition to long-term permafrost monitoring at the Research
Station Samoylov Island (Hubberten et al., 2006; Boike et al., 2019), extensive records on
meteorology, soil and ecosystem characteristics (Zibulski et al., 2016; Boike et al., 2019; Boike et al.,
2008), hydrology (Fedorova et al., 2015), and greenhouse gas fluxes (Rossger et al., 2022; Holl et
al., 2019) are available, setting an important benchmark for further assessments of changes in an
Arctic river delta. During the summer season of 2018, an extensive field campaign to the Lena Delta
led to an unprecedented amount of field datasets including vegetation cover recordings, above
ground biomass estimates, and spectral characterisation of the different vegetation/land cover units.
These in situ datasets provide improved thematic detail allowing the development of habitat
classifications. In 2009, Schneider et al. (2009) developed the first land cover classification map for
the entire delta at 30 m spatial resolution based on Landsat-7 ETM+ satellite summer images from
2000 and 2001 to quantify delta-wide methane emissions. The availability of Sentinel-2 (Sentinel-2)
Multispectral Instrument (MSI) data from two orbiting satellite missions since 2016 and 2017 provide
high quality multispectral satellite data with a higher spatial resolution in the Visible and Near
Infrared wavelength of up to 10 m, and of 20 m in the Red Edge and the Short- Wave Infrared
wavelength regions (Drusch et al., 2012, ESA 2015). Together with the extensive ground
observations from the Lena Delta in 2018 this enables an updated classification, using the higher
resolution Sentinel-2 images and improved thematic detail.
In the following study, field datasets as well as derived multispectral satellite images from the
summer season 2018 for the Lena Delta were used to provide 1) an updated data-driven framework
for plant communities and associated habitat classes in the Lena Delta, 2) a high-resolution habitat
mapping product for the entire delta, and 3) a disturbance regime map linked to habitat classes.
These datasets enhance our understanding of the Lena Delta system and will build a baseline and
framework for future spatio-temporal analysis of more detailed processes and changes within this
highly sensitive ecosystem.

# 117  2 Study Area

The Lena Delta is located in northeastern Siberia's continuous permafrost zone between 72° and
74°N and 123° to 130°E (Figure 1). With an area of about 30 000 km$^2$, it is the largest delta in the
Arctic and one of the largest in the world (Walker, 1998; Schneider et al., 2009). It is surrounded by
the Laptev Sea to the west, north, and east, and the Chekanovsky and Kharaulakh mountain ranges
border it to the south. The delta is characterized by numerous river channels and more than 1500
islands with a diverse geologic history (Grigoriev, 1993). Morphologically, the delta can be divided
into three distinct geomorphological main terraces (Grigoriev, 1993; Schwamborn et al., 2002). The
first main terrace, which comprises the Holocene fluvial terraces and the active floodplains, is the
youngest and most active part of the delta (Schwamborn et al., 2023), and covers most of the east-
northeastern areas as well as the southern and southwestern-most parts This main terrace
predominantly consists of ice wedge-polygonal tundra (Nitzbon et al., 2020) as well as of barren and
vegetated floodplain areas (e.g., Rossger et al., 2022). The second main terrace, located in the
northwestern part, contains mostly sandy, comparably well-drained soils with low ground-ice content
(Schwamborn et al., 2002; Ulrich et al., 2009). Large, mostly north-to-south oriented lakes and
depressions are abundant in this area (Morgenstern et al., 2008). The third and oldest main terrace
consists mainly of remnants of a Late Pleistocene accumulation plain with ice- and organic-rich
sediments (so-called Yedoma deposits) and is characterized by polygonal tundra with large ice
wedges, deep thermokarst lake basins, and thermo-erosional valleys (Morgenstern et al., 2011;
Morgenstern et al., 2021). The third terrace is found on islands in the southern delta region
(Schirrmeister et al., 2003; Schirrmeister et al., 2011). Permafrost in the area has a thickness of
about 500–600 m (Romanovskii and Hubberten, 2001). The active layer depth, i.e., the seasonally
thawing upper soil layer, on the first terrace is usually in the range of 30 to 50 cm and 80 to 120 cm
on the floodplains (Boike et al., 2019). The larger region is characterized by an Arctic continental
climate with low mean annual air temperatures of −13 °C, a mean temperature in January of −32 °C,
and a mean temperature in July of 6.5 °C. The mean annual precipitation is low and amounts to
about 190 mm (World Weather Information Service).
As part of past Russian-German expeditions to the Lena Delta, most research during the last two
decades has been carried out on the islands of Samoylov and Kurungnakh in the central delta
(Figure 1). Samoylov Island (72°22′ N, 126°29′ E) covers an area of about 5 km$^2$ and is
representative of the first terrace together with an active floodplain (Boike et al., 2019; Boike et al.,
2008). The vegetation and soil types are diverse at local scales due to high lateral variability of the
polygonal microrelief consisting of drier polygon rims, and moist to wet polygonal depressions and
troughs (Nitzbon et al., 2020; Kienast and Tsherkasova, 2001). In contrast, Kurungnakh Island is
mainly composed of late Pleistocene Yedoma deposits that belong to the third delta terrace
(Grigoriev, 1993) with elevation up to 55 m above sea level (m a.s.l.) (Morgenstern et al., 2013).
Holocene cover deposits and peat-rich permafrost soils are distributed across the surface of the third
Lena River terrace and especially concentrated in the deep thermokarst basins called "alases".
Alases are important landscape-forming features of the ice-rich Yedoma permafrost zone, which are
mainly caused by extensive melting of excess ground ice in the underlying permafrost (Van
Everdingen, 1998).

# 159 3 Datasets and methods

Several new datasets are presented for the Lena Delta that are spatially and thematically connected
and support vegetation, habitat, and land cover applications for this region (Figure 1).
Two datasets feature field-measured vegetation data, providing information on foliage projective
cover (Dataset 1) and above ground biomass (Dataset 2) recorded in the central Lena Delta in
summer 2018 across 26 selected vegetation plot sites (supplementary Table S1, S2). The field plots
of 30 x 30 m (900 m$^2$) were chosen to be representative for typical vegetation communities (vascular
plants, moss and lichen cover) as largely homogenous sites representative for the surrounding area.
In addition, a total of 28 in-situ, canopy-level hyperspectral field measurements were acquired in 30
x 30 m plots with homogeneous vegetation or barren to partially vegetated areas (spectral
reflectance field measurements; Dataset 3). Of the 28 hyperspectral measurements, 15 were
conducted at the vegetation plot sites of Datasets 1,2 three measurements were repeat
measurements to capture vegetation senescence, and at 10 spectrometry plots we conducted
hyperspectral field measurements without floristic inventories but with detailed plot documentation.
Based on expert knowledge, we defined representative habitat classes and identified homogeneous
regions within the central Lena Delta to train and apply a classifier using a Sentinel-2 satellite image
from summer 2018 (Dataset 4). Due to the high reliability of the central Lena Delta vegetation
classification and positive evaluation by field experts, we used this vegetation classification as a
training dataset for a robust classifier that was subsequently applied to a Sentinel-2 image mosaic
for the entire Lena Delta for 2018 to develop a new Lena Delta habitat map (Dataset 5).
Finally, using the habitat classes, probability maps for exposed sandbars and water distribution, and
information from the in-situ dataset (Datasets 1 & 2), we extrapolated a classification of disturbance
regimes across the delta (Dataset 6) as an application example for the habitat classes.
**3.1 Foliage projective cover (Dataset 1)**
A detailed description of plant composition for the 26 vegetation plots of the 2018 expedition to the
Lena Delta was compiled (see supplementary Tables S1-3). Prior to the field work, the approximate
site locations were defined for establishing representative vegetation plots based on field knowledge
and evaluation of Landsat and Sentinel-2 satellite imagery. The aim was to cover representative
vegetation communities of the central delta. There are vegetation communities with large area
coverage that show high homogeneity within larger areas (10s of meters). Therefore, at each site
location, we defined a 30 x 30 m square plot with a homogeneous or repetitive vegetation
composition that was also representative of the wider land surface serving as an Elementary
Sampling Unit (ESU). ESUs according to the Committee on Earth Observing Satellites Working
Group on Calibration and Validation (Duncanson et al., 2021) serve as spatial training and
validation units representative for the land surface for quantitative and qualitative remote
sensing operations. In case of more patchy and heterogeneous vegetation structure we selected 30
x 30 m squares embedded in a minimum of 50 x 50 m square of the same vegetation composition.
The detailed floristic composition was recorded around the plot center in four successive rings of 50
cm diameter. In addition, the vegetation plot was mapped in detail from above with one Red-Green-
Blue (RGB) and one Red-Green-Near Infrared (RGNIR) MAPIR camera using telescope stick-based
field photography. The projective vegetation cover was recorded in at least three subplots (2 m x 2
m)  within the plot. If the vegetation cover was highly homogenous three subplots were established.
In the case of moisture differences, e.g. in polygonal tundra with dry rims and moist to wet
depressions, we established higher numbers of subplots capturing moist as well as dry patches
(see, Figure 2 & 3 describing the concept). We compiled the floristic composition to foliage projective
cover by plant taxa on each 2 x 2 m subplot for the different canopy levels and extrapolated for the
30 m x 30 m plot. We used the RGB and NIR field photos to make an estimate on the share of moist
and dry surface area to calculate an averaged projective vegetation cover. The ring survey data was
not included in the plot average. The dataset of percentage foliage projective cover per vegetation
plot is published in PANGAEA (Shevtsova et al., 2021a,
https://doi.pangaea.de/10.1594/PANGAEA.935875).

## 3.2 Above ground plant biomass (Dataset 2)

Above-ground biomass (ABG) was sampled in the field in 25 of the 26 vegetation plots in 2018 (see
supplementary Tables S1-3). Within each 2 x 2 m subplot a 0.5 m x 0.5 m representative plot was
selected for ABG sampling. AGB sampling for moss and lichens was conducted within 0.1 m x 0.1 m
subplots inside the 0.5 m x 0.5 m subplots.
In total, 174 fresh AGB samples were collected and weighed in the field or subsequently at the
Samoylov research station. AGB samples with a weight exceeding 15 g were subsampled. The plant
samples were then dried for two to four days in a warm dry place and finally oven-dried for ca. 24
hours at a temperature of 60 °C before re-weighing. All AGB assessments per plant community type
were upscaled to the 30 m x 30 m plot in $g/m^2$ using the foliage projective cover data. The dataset of
AGB per vegetation plot has been published in PANGAEA (Shevtsova et al., 2021b,
https://doi.pangaea.de/10.1594/PANGAEA.935923).

## 3.3 Hyperspectral field measurements (Dataset 3)

Hyperspectral field measurements were conducted in the central Lena Delta in August 2018 with the
aim to collect surface reflectance spectra of different homogeneous land cover units across a variety
of delta land surfaces and vegetation composition. In total, we collected 28 hyperspectral field
measurements in homogeneous 30 x 30 m spectrometry plots (Table S5), with 15 of them equalling
the vegetation plots across Samoylov and Kurungnakh islands (see Dataset 1 & 2 and
supplementary Table S4), three as repeat measurements at the end of August to capture the change
in spectral signature during senescence since the beginning of August and the remaining 10 field-
spectroscopy plots focusing on non-vegetated areas such as sandy parts of the floodplain. We
conducted the field-spectroscopy measurements with a Spectral Evolution SR-2500 field
spectrometer with a 1.5 m Fiber Optic Cable. The instrument was calibrated to spectral radiance
within a wavelength range of 350 to 2500 nm. Within the 30 x 30 m homogeneous spectrometry
plots we acquired about 100 individual measurements, randomly scattered across the plot. Before
and after each survey we conducted reference measurements by measuring the back reflected
downwelling radiance from a Zenith Lite[TM] Diffuse Reflectance Target of 50% reflectivity to normalize
to surface reflectance percentages per wavelength. The averaged individual measurements of the
reflectance of each spectrometry plot was published in the PANGAEA data repository (Runge et al.,
2022, https://doi.pangaea.de/10.1594/PANGAEA.945982).
**3.4 Central Lena Delta habitat classification (Dataset 4)**
*3.4.1 Habitat classes*
Based on the vegetation plots (Dataset 1 & 2) and from field knowledge, different habitat classes
characterized by distinct plant communities, moisture regimes and soil properties were defined. Non-
vegetated areas (e.g., sand) and water were added as additional classes using band thresholds
(Table 1). During an iterative process within a Sentinel-2 based supervised classification, additional
habitat classes that were not covered by the vegetation plots (Dataset 1 & 2) were added: i) The
polygonal tundra complex could spectrally be separated into distinct classes related to different
surface water abundance in the form of intra- and interpolygonal ponds, therefore, we implemented
three different polygonal tundra complex classes, with up to 10%, 20%, 50% surface water cover
respectively, and ii) one class of 'sparsely vegetated' representing the areas of transition zones
between vegetated and barren. Table 1 provides details on habitat class descriptions and
established methods to distinguish habitats.
*3.4.2 Satellite data processing*
The central Lena Delta habitat classification is based on one high quality cloudless Sentinel-2)
image from August 6 in 2018, representing the late summer. The Sentinel-2 top of atmosphere
reflectance (TOA) image data was processed by the German Space Agency DLR (B. Pflug, oral
communication, 2019) to bottom of atmosphere (BOA) surface reflectance using the newest version
of the atmospheric correction processor Sen2Cor later released as ESA Sen2Cor in 2020.
Atmospheric correction processing was performed with the default rural aerosol model. All spectral
bands were resampled to the 10 m pixel resolution bands. The 60 m pixel resolution bands (B1, B9,
B10) that support atmospheric correction, but are not optimal for land surface classification, were
removed. We added the normalized difference vegetation index (NDVI; NIR-RED / NIR + RED) to
the band collection.
*3.4.3 Central delta habitat classification*
Sentinel-2 pixels from the 30 x 30 m ESUs (dataset 1, Shevtsova et al. 2021a), and additional
polygonal shapefiles (Figure A3) defined by expert knowledge, led to a training dataset of 8 626
labelled pixels for the habitat classification (labelled pixels are published in the Landgraf et al 2022a
data collection). An independent test dataset of polygonal shapefiles with habitat annotation was
delineated based on high resolution satellite and drone images, S-2 NDVI and SWIR bands and in
areas that have been visited regularly during field expeditions (Figure A4).
From the training dataset we randomly selected  4 313 pixels to train the classifier. We tested
several classifiers and different selected band combinations (spectral bands and NDVI). Water
(transparent to turbid) and sandbanks were omitted in the classification processing by masking them
as inactive using a band threshold; the water mask was based on the NIR 10 m band 8 (NIR < 0.02)
and the sand mask was based on the blue 10 m band 2 (Blue > 0.07, Table 1). The classification
was tuned to depict vegetation composition and was qualitatively assessed well known to the
classification developers. Best results for the habitat classification were obtained using a random
forest classification with a band combination of all Sentinel-2 VIS, Red-Edge, NIR and SWIR bands,
and the NDVI. The chosen classifier was able to distinguish between relevant classes (Table 1) and
could even identify patchy spots of specific habitat classes. In addition to the defined water and sand
classes, the final central Lena Delta classification contains 10 habitat classes (Table 1). The here
defined central Lena Delta covers an area of 644.9 km$^2$ with a 55.2 % vegetation cover.
To assess the classification performance, we applied a cross-validation on a random selection of
locations within the independent test dataset and used landscape descriptions at permafrost coring
sites (Siewert et al. 2016 a,b,c) (Figure S 6). We used 34 locations that we could relate to categories
such as polygonal tundra, wetlands, and sandy areas. These broad land cover categories matched
well (Table S8).  For the evaluation, 100 random points per pre-defined habitat class were selected
from the test dataset. Based on a confusion matrix, the overall classification accuracy was 94.00 %
(class-based accuracy and statistics shown in Table A1). More importantly, the accuracy was
qualitatively tuned and evaluated based on ground-truthed knowledge of the development team. The
published dataset of Landgraf et al. (2022, https://doi.pangaea.de/10.1594/PANGAEA.945057)
provides the central Lena Delta habitat classification map, the ESUs and the polygons used to train
the classifier. The training dataset includes data from 23 of the 26 vegetation plots (dataset 1). The
dataset provides additional 69 ESUs defined with expert knowledge gathered during several field
expeditions to the Lena Delta, labeled as pseudo ESUs for potential future investigations.
**3.5 Lena Delta habitat classification (Dataset 5)**
*3.5.1 Lena Delta habitat classes*
In order to extend the habitat classification map to the entire Lena Delta (29873.7 km$^2$), we included
all the habitat classes covering the central Lena Delta (dataset 4, Table 1). In addition, and based on
expert knowledge as well as extensive visual satellite image investigations, we added one habitat
class that is not present in the central Lena Delta: the second terrace in the northwest of the Lena
Delta is lithologically and geomorphologically different from the other two terraces present in the
central delta, and characterized by sandy substrates. In a hyperspectral CHRIS PROBA satellite-
based geomorphological classification, Ulrich et al. (2009) described the second terrace featuring
very dry elevated sandbanks, barren or poorly vegetated areas with isolated lichens, moss, herbs,
dwarf shrubs or grasses (vegetation cover 0–60%, growth height: max. 20 cm, average active layer
depth of 1 m on the upland plain with old, vegetation-arrested sand dunes). Based on photos taken
at few locations in the field during past expeditions (see supplementary table S3) the habitat class
shows well-drained areas dominated by sandy substrate and diverse, sparse vegetation cover; some
areas are dominated by sedges, cotton grass and mosses with rare occurrences of lichens and
dwarf shrubs, while some areas are dominated by the latter. Schneider et al. (2009) defined the
same class as 'dry moss-, sedge- and dwarf shrub-dominated tundra (DMSD)'. We selected 35
ESUs for this habitat class characterized by high SWIR reflectance (Sentinel-2 band 11) due to dry
land surface conditions. The habitat class was named 'dwarf shrub - herb communities' and was
added as an additional habitat class to the training data set.
*3.5.2 Satellite data processing*
The Lena Delta habitat classification was based on a Sentinel-2 mosaic (top of atmosphere (TOA)
reflectance, Google Earth Engine Dataset) with images taken of the area between June 1 and
September 15, 2018. The images (N = 1685, distributed across 15 Sentinel-2 tiles) were filtered to
discard images with cloud cover above 20%. A cloud mask was applied to the remaining 262
images, masking pixels where the quality band 'QA60' indicates clouds (band 10) or cirrus (band
11). All spectral bands with 20 m resolution were resampled to match the 10 m resolution bands.
Next, NDVI was computed (see 3.4) for each image and one high-quality mosaic of all images based
on the maximum NDVI value per pixel was produced representing a snapshot of the peak summer
vegetation period. Using the median NIR band values across the 262 cloud-masked images, we
classified water with a threshold of < 0.07 reflectance. The remaining non-vegetated areas defined
by a threshold of NDVI < 0.4 were classified as barren/sand. The water- and sand-masked image
mosaics were then used in the classification pipeline with the following bands: B2 (blue), B3 (green),
B4 (red), B5 (red edge 1), B6 (red edge 2), B7 (red edge 3), B8 (NIR), B11 (SWIR 1), B12 (SWIR 2),
and NDVI.
*3.5.3 Lena Delta Habitat classification*
From the central Lena Delta habitat classification (dataset 4) we sampled 7 500 random pixels to
train a random forest classifier (smileRandomForest in Google Earth Engine). In addition, we added
35 pixels from the ESUs selected within the 'dwarf shrub - herb communities' of the north-western
Lena Delta. Given the dominance of the 'dwarf shrub – herb communities' on the second terrace
(north-eastern part of the Lena Delta), the confidence of selecting correct training pixels for this
habitat was relatively high (see also Figure S7). Unfortunately, no vegetation recording or monitoring
schemes exist outside the central Lena Delta. The accuracy of the classification was quantified using
the independently defined shapefiles  within the central Lena Delta (same dataset used to quantify
the accuracy of the central Lena Delta habitat classification, Figure A4 and Table S1). Based on a
confusion matrix, the overall classification accuracy was 85.06 % (class-based accuracy and
statistics shown in Table A2). Similar to the validation of the central Lena Delta habitat classification,
the results were carefully checked to make sure that large-scale pattern, e.g., differences between
the three terraces, are accurately separated, and that the highly repetitive structures within terraces
are also recognized by the classification (see Figures S6-S8).
Since the barren/sandy areas are highly dynamic with variable water levels mainly within (due to
flooding in spring and decreasing river flow during the summer season) but also across years
(discharge dynamics), we computed a sandbar probability map for the Lena Delta using cloud
masked Sentinel-2 (TOA reflectance) images between April 1 and October 15 from 2015 to 2021 (6
026 images). In each image, we labeled sandy pixels by NDVI < 0.4 AND NDWI > 0.095 AND NIR <
0.09 reflectance. Next, for each pixel in the Lena Delta, we computed the percentage of sandy pixels
across all images resulting in a sand probability map. The training dataset (random 7500 points, plus
35 points with label 'dwarf shrubs - herb communities'), the habitat classification, and the sand
probability map was published in the PANGAEA repository (Figure 5, Lisovski et al., 2022,
https://doi.pangaea.de/10.1594/PANGAEA.946407).
**3.6 Lena Delta disturbance regimes (Dataset 6)**
The Lena Delta experiences different disturbance regimes, mapped and described in dataset 6.
Mainly annual flooding, but also local rapid thaw processes on the land surface of the terraces with
ice-rich permafrost, result in disturbance regimes forming distinct habitat classes (Table 2). The
floodplains experience seasonal flooding as a regularly occurring disturbance in spring after ice-
break up (the spring flood). Very high disturbance regimes due to the most intense scour, erosion
and sedimentation result in barren sandbanks or in early-stage plant communities equalling the
'sparsely vegetated' habitat class. The classes 'moist to wet sedge communities', 'wet sedge
communities', 'moist equisetum and shrubs', 'dry shrub communities', 'dry grass to wet sedge
communities' represent the mid to advanced successional stages on the floodplain within areas of
high disturbance that are also described as shifting habitat class (Stanford et al., 2005; Driscoll and
Hauer, 2019).
In contrast to the high disturbance regimes on the floodplain, habitats on the first, second and third
delta terraces are less extensively disturbed (low disturbance). In these areas typical mature-state
tundra plant communities are able to develop; 'polygonal tundra complex', 'tussock tundra', and
'dwarf shrub herb communities'. However, locally, high disturbance occurs by rapid thaw processes
of ice-rich permafrost on the first and third delta terraces with habitats characterized by mid to
advanced-stage plant succession; 'moist to wet sedge communities', 'wet sedge communities', 'dry
shrub communities', and 'dry grass to wet sedge' communities. Very high disturbance due to intense
rapid thaw processes occurs at eroding cliffs and lake margins, in steep valleys and actively
developing gullies resulting in barren surfaces with rims of sparsely vegetated transition zones.
Given the link between plant communities and flooding as well as rapid thaw processes, we
characterized the disturbance regimes for each habitat class (Table 2) and provide mapped
disturbance  based on the habitat class of dataset 5 and the corresponding disturbance regime for
the entire Lena Delta (Figure 6, Heim and Lisovski, 2023, https://doi.org/10.5281/zenodo.7575691).

# 382  4 Results and Discussion

We deliver a detailed description and associated data products of the most prominent habitat
classes in the largest Arctic river delta, the Lena Delta. Supported by ecological field data of plant
composition, hyperspectral field measurements from the same sites, and regional expert knowledge
collected over decades, we develop a high-resolution Sentinel-2 based habitat map for the entire
delta. The compiled datasets provide the necessary baseline for future investigations of the
biochemical processes, ecological dynamics, and responses to global warming within the Arctic
tundra system of the delta.
**4.1 Habitat classes of the Lena Delta**
Based on the floristic composition and biomass of the vegetation plots (Dataset 1, 2), the spectral
properties from hyperspectral field measurements (Dataset 3) as well as expert knowledge, we
defined 11 distinct habitat classes linked to different vegetation composition for the Lena Delta
(Figure 4). The selected Sentinel-2 spectral bands and the derived NDVI values allow a separation
of the habitat classes into two distinct groups (the first separation level between habitat classes in
Figure 4a, 1st hierarchical level). Three habitat classes ('wet sedge communities', 'moist Equisetum
and shrub communities', 'dry grass to wet sedge communities') formed in areas of high disturbance
by rapid thaw processes and regular flooding represent a distinct cluster with highest vegetation
vitality (high NDVI), and separated from the more stable and mature tundra communities ('polygonal
tundra complex', 'dry (tussock) tundra', and 'dry dwarf-shrub and herb communities'), and the other
successional plant communities ('moist to wet sedge complex', 'dry low shrub communities' and
'sparsely vegetated') all characterised by a lower NDVI range. The 'dry dwarf-shrub and herb
communities' form a separate cluster with the least overlap with other habitat classes within the two-
dimensional non-metric multidimensional scaling (NMDS) space (2nd hierarchical level, Figure 3a;
Figure 4c) due to very low vegetation vitality and surface moisture (lowest NDVI, high red and SWIR
reflectance). There are two remaining habitat classes on the 3rd and 4th hierarchical level, which are
successional plant communities, the *'moist to wet sedge complex'* and 'dry low shrub communities'.
The separation on the 3rd and 4th hierarchical level is mainly driven by higher NDVI of these
successional plant community classes in comparison with the mature state tundra plant communities
with lower NDVI (Figure 4a-b). The 'dry grass to wet sedge communities' and the 'sparsely
vegetated area' habitat class (not covered by vegetation plots but added during the classification
process), show the largest overlap with the other habitat classes due to a high variability in
vegetation cover, biomass and moisture. In general, the ordination method (Figure 4b) shows that
distinct plant communities and the associated habitat classes are mostly separated by a biomass
gradient for which the NDVI is a good approximator. A further separation linked to potential spectral
proxies for biomass exists with the far red-edge and NIR bands (B6,7,8) but is less distinct than the
NDVI axis. Together with the SWIR (B11,12) the red (B4) and near red-edge (B5) bands, and less
strongly the blue and green bands (B2,3), the results indicate a habitat class separation based on
moisture, biomass and vegetation colour characteristics.
The vegetation plot selection was made in relation to the most typical habitats (e.g., Mueller-
Bombois and Ellenberg, 1974). For 15 of the 26 vegetation plots, we collected and provided
hyperspectral surface reflectance data (Runge et al., 2021). These measurements cover a variety of
landscape units including Yedoma uplands, floodplains (vegetated and non-vegetated), drained
thermokarst lake basins (old and recently drained), and areas covered by low shrub layers.
Comparing the hyperspectral surface reflectance with multispectral Sentinel-2 data, we found
commonalities in the discrimination of habitat classes along moisture gradients. Unfortunately, the
hyperspectral field measurements do not cover the biomass gradient. Plot measurements with the
field spectrometer are conducted with the hand-held instrument held at shoulder height, hence it was
not possible to acquire field spectroscopy measurements in disturbed patches with tall shrubs or
very sloped terrain. This highlights the difficulty in deriving high spectral resolution surface
reflectance measurements representative of fine scale differences between Arctic tundra habitat
classes if the plot properties become too challenging to measure.
In general, mature-state tundra plant communities have relatively similar spectral properties due to
low vascular plant cover (e.g., Beamish et al., 2017). In addition, the tundra vegetation communities
contain a wide range of accessory pigment composition (carotenoids and anthocyanins) that result in
a very similar spectral response (Beamish et al., 2018). Only the highly disturbed communities such
as wetlands or areas with tall shrubs are more spectrally distinct due to a high NIR reflectance
plateau (Buchhorn et al., 2013). Since the hyperspectral field measurements provide a higher spatial
resolution and thus also a measure of variability within areas of the same general habitat type, we
consider the measurements valuable for applications that aim at analysing ecological and
biochemical processes within distinct habitats in more detail.
**4.2 Sentinel-2 based habitat classification**
Based on the identified habitat classes (Table 1) we applied a random forest classifier to map habitat
classes in the central Lena Delta and subsequently in the entire Lena Delta. Both maps represent
the summer season of 2018 for which we could use a sufficient number of satellite images with low
cloud cover.
The Lena Delta habitat map shows the ice-rich first and third terraces mainly covered by i) the
'polygonal tundra complex' due to impeded drainage on the terrace plateaus and by ii) drier tundra
communities on well drained areas due to older degraded permafrost forms (detailed description in
Morgenstern et al., 2008, 2011). On the second terrace, the classified 'dry dwarf shrub and herb
communities' occur well separated from the moist habitat classes covering the floor of the alases.
On the floodplains, the rich mosaic outlines a wide spectrum of very diverse classes, the dry versus
moist and wet substrate habitats, in the active delta area.
Polygonal tundra is characterized by high spatial heterogeneity; at the decimeter to meter-scale
plant composition and diversity is defined by the polygonal microrelief and water level (Whitaker and
Woodwell, 1968; Forman and Godron, 1981; Zibulski et al., 2016; Nitzbon et al., 2020, Siewert et al.,
2021). Therefore, within a single Sentinel-2 pixel, dry polygonal rims, moist slopes, wet patches and
surface water can all be present. The spatial resolution of Sentinel-2 cannot capture the meter-scale,
but captures the heterogeneity between the different surface water contributions of the 'polygonal
tundra complex' on the first and third terrace. In the Lena Delta, the 'polygonal tundra complex with
up to 50% surface water' represents the dominant habitat class with 25% of the delta area (about 7
434 km$^2$). All other habitat classes represent 1-6% of the delta area with 'dwarf shrub-herb
communities' and 'moist to wet sedge complex' reaching 5.4% and 5.9%, respectively (Figure 6).
Based on the summer Sentinel-2 mosaic, the classes 'Water' and 'Sand' cover more than 40% of the
delta. However, those two classes are extremely variable within and across years, depending on the
river water level during image acquisition time. To provide information on this variability, we
calculated how often each pixel in the delta (cloud free Sentinel-2 pixels from 2015 to 2022) was
classified as sand (threshold approach). This led to an additional sand probability layer with values
between 0-100%.
Despite extensive research within the area, only a few classification products are available for the
Lena Delta. The new Lena Delta classification is a high-resolution (Sentinel-2, 10 m) map that
focuses on the delta-specific habitat classes and emphasizes the high level of heterogeneity across
the delta. We compared the Lena Delta habitat classification to existing classifications: the first
published Lena Delta-wide land cover classification targeted towards tundra environments and the
upscaling of methane emissions with 30 m resolution (Schneider et al., 2009), the global ESA
Climate Change Initiative CCI land cover classification with 300 m resolution (Defourny, 2019), and a
circum-arctic standardized ESA GlobPermafrost land cover map of the Lena Delta with 20 m
resolution (Bartsch et al., 2019). We sampled the classification results with a regular point grid of
more than 3 million points which have an equal distance of 100 m to one another to compare the
classification results. Figures and tables with more information on class comparisons can be found in
the supplements (Table 1, Figure S3-5). Overall, the classifications of the Lena Delta overlap well for
'water' (water bodies (Defourny, 2019), shallow water (Schneider et al., 2009), water (different
depths and sediment yields, Bartsch et al. 2019)) and 'sand' (bare areas (Defourny, 2019), mainly
non-vegetated areas (Schneider et al., 2009), sand, seasonally inundated and disturbed (Bartsch et
al. 2019)) areas. Besides this, the mapped classes differ greatly from one another. For example, the
dominant classes in the coarse ESA CCI land cover 2018 product (300 m) for the Lena Delta are
'shrub or herbaceous cover', 'flooded', 'fresh / saline / brackish water', 'sparse vegetation (tree,
shrub, herbaceous cover) (<15%)', and 'mosaic tree and shrub (>50%)', 'herbaceous cover (>50%)'.
These broad classes describe the major land cover in the Arctic delta but fail to depict the
heterogeneity of habitats and plant communities not only because of its coarse spatial resolution but
also because of the broad class descriptions. Furthermore, smaller areas are classified as 'tree
cover', 'needleleaved', 'evergreen / deciduous', 'closed to open (>15%)' and 'mosaic tree and shrub
(>50%) / herbaceous cover (<50%)' which is an inaccurate depiction of the delta.
This habitat map and the land cover classification from Schneider et al. (2009) resemble each other
more closely, however, this habitat map shows more differentiation in the classes and spatial
resolution, 10 m to 30 m, respectively. The only class description that is identical in both
classifications, besides water and sand / mainly non-vegetated areas, is 'dry tussock tundra'.
However, there is only a small match between these classes in the point comparison and most 'dry
tussock tundra' areas from the Schneider et al. (2009) classification fall into the PC_50%:, PC_20%,
'moist wet sedge complex' and 'dwarf shrub-herb communities'. The habitat map shows the mosaic
of habitats on the floodplain with 'moist equisetum and shrubs on floodplain', 'dry low shrub
community', 'moist to wet sedge' and 'wet sedge complex' which match with 'moist to dry dwarf
shrub-dominated tundra' in the land cover classification of Schneider et al. (2009). Also, for the
polygonal tundra complex, our habitat map shows more differentiation with three classes of up to
50% 20% 10% surface water contribution versus two classes in Schneider et al. (2009) 'wet sedge
and moss dominated tundra' and 'moist grass and moss dominated tundra' The areas covered by
'PC_50%' and 'PC_20%' match with 'wet sedge- and moss-dominated tundra', and 'PC_20%' and
'PC_10%' match with 'moist grass and moss-dominated tundra'. The overall aim of both maps is to
differentiate between dry to wet land cover habitats as these describe the heterogeneity in the delta
well and determine factors related to methane emissions (see Schneider et al. 2009) and the
different habitat classes.
The land cover classification from ESA GlobPermafrost differentiates between 21 classes which are
associated to eight broader groups, such as sparse vegetation, shrub tundra, forest, grassland,
floodplain, disturbed, barren and water (Bartsch et al., 2019). With a spatial resolution of 20 m, the
latter product is the closest to this habitat map. The major class 'wet ecotopes' of ESA
GlobPermafrost match with our 'PC_50%:' on the first terrace and the 'moist to wet sedge complex'
on the floodplains. On the floodplain however, other classes show less agreement. The ESA
GlobPermafrost one class 'floodplain mostly fluvial' does not differentiate the floodplain classes
further, in contrast to our habitat map differentiating between 'moist to wet sedge complex', 'wet
sedge complex', 'moist equisetum and shrubs' and 'dry low shrub community' on floodplain.
Whereas the ESA GlobPermafrost class 'disturbed' (defined as forest fire scars, seasonally
inundation and landslide scars can be found in 'PC_50%' predominantly, in 'sand', 'PC_20%' and
'sparsely vegetated areas' in our habitat map. This underlines the complex structure of match and
mismatch between classifications.
The land cover map from Schneider et al. (2009) is based on two cloud-free Landsat images from
June/July 2000 and 2001, the ESA CCI land cover 2018 map is based on summer images as well.
Hence, the images used for this habitat classification were acquired at a similar time as for the ESA
CCI product and we do not expect differences based on changes on the ground due to this temporal
concurrence. In the almost 20-year difference between Schneider et al. (2009) and this habitat map
we do expect changes in vegetation composition. Overall, it is challenging to obtain sufficient cloud-
free images during the summer months to fully cover the entire Lena Delta for a classification project
and to depict a specific phenological state. Therefore, we created a Sentinel-2 composite mosaic
based on the maximum NDVI value per pixel from June to September. With this we ensure to have
the peak vegetation and phenology season represented as input for the habitat classification as
much as possible and increase comparability to other classification studies despite a temporal
mismatch.
The habitat map gives an accurate and detailed description of the Arctic Lena Delta that
incorporates extensive field data and expert knowledge. The habitat map is superior to the ESA CCI
land cover map (2018) in both spatial resolution and class description as it depicts the
heterogeneous habitat distribution. The 20m ESA GlobPermafrost classification matches the
resolution of the habitat map closely but due to its wider geographical application with circum-Arctic
standardized classes it does not optimally represent Lena Delta-specific habitats, such as the widely
distributed polygonal tundra complex. Furthermore, the habitat map is an update to Schneider et al.
(2009), which was based on three Landsat images from 2000 and 2001 and shows further
differentiation of habitats, specifically representing the floodplain mosaics of this Arctic delta.
**4.3 Habitat linked disturbance regimes**
Parts of the Lena Delta are characterised by disturbances due to annual floodings or rapid
permafrost thaw processes leading to specific habitat classes. We provide habitat linked disturbance
regimes (describing the type and intensity of disturbances) across the delta. Our product (Dataset 6,
Figure 6a) shows that the largest part of the vegetated delta (excluding 12 439 km$^2$ of 'sand' and
'water' classes) is impacted by low disturbance, resulting in mature-state plant communities on the
terrace plateaus (Figure 6b, 72%, 12 806 km$^2$). Specifically, the second terrace in the northwest of
the delta, with low ice content, is least impacted by rapid thaw processes and not part of the active
delta. In contrast, the habitats in the active delta are all linked to high disturbance (27%, 4 875 km$^2$).
The 'moist to wet sedge complex' (10% of the vegetated Lena Delta) is the largest class considered
to be formed by high disturbance. This class is found in larger patch sizes on the riverine floodplains,
smaller patches on the floor of thermo-erosional valleys. Overall, 27.5% of the vegetated area of the
Lena Delta experiences some level of high disturbance from either regular spring floods or from
rapid thaw processes.
Species richness, relative abundance and biomass characteristics are important habitat features that
are influenced by landscape characteristics such as topography, water fluxes, soil types and
disturbance regimes (Forman and Godron, 1981; Naiman et al., 1986; Pickett et al., 1989;
Montgomery, 1999). Greig-Smith (1964), Woodwell and Whittaker (1968), and Forman and Godron
(1981) described fragmentation of land surfaces due to disturbance (defined by type and intensities)
and topography. In the Lena Delta, the terrace-related topography and active floodplain areas are
major determinants of plant communities and habitat classes and are thus well reflected in the Lena
Delta habitat map.
The high disturbance regime on floodplains results in 'shifting habitats' (Stanford et al., 2005; Driscoll
and Hauer, 2019). The annual spring floods and rapid thaw processes result in areas of high
disturbances, habitats of mid to advanced plant successional stages showing high vascular plant
above ground biomass (Figure 6c) due to the higher nutrient availability, a deeper active layer and
more moisture (e.g., Myers-Smith et al., 2020). Within the low disturbance habitat classes, a thick
moss layer as well as a low vascular plant coverage characterise the tundra community
assemblages representing mature state plant communities. Because high disturbance patches are
characterized by high vascular biomass, they can be well classified specifically in the NDVI, but also
NIR and red edge bands of optical medium resolution sensors such as SENTINEL-2. Within the
vegetation plots (Dataset 1), we did not find clear differences in species richness and in the Shannon
diversity index between the disturbed and the undisturbed classes (Figure 6d). Since most disturbed
habitat classes such as the 'moist to wet sedge', the 'wet sedge' as well as homogeneous patches of
high shrubs (as part of the habitat class 'dry grass to wet sedge complex'), were not sampled in the
field due to too challenging conditions, however they are clearly representing habitats with low
species richness. In the extreme case disturbance can lead to barren and sparsely vegetated
surfaces.
**4.4 Classification accuracy and representativeness**
The field data was acquired during a field trip in July-August 2018, primarily focusing on 30 m x 30 m
homogeneous vegetation and land cover plots. Additionally, we relied on Sentinel-2 images for the
different classifications that were also acquired in summer 2018, covering the same period as the
field trip, and have a spatial resolution of 20 m. The temporal overlap of the field work and the
satellite image acquisitions ensures consistency across the different datasets and represents a close
relationship between datasets and products obtained in the field (dataset 1, 2 and 3) and the results
derived from the satellite images that use the field data as input. As Sentinel-2 images have a small
geolocation error, we could link our field plot locations directly with the satellite images. Furthermore,
the sampling and measurement design of the plots with 30 m x 30 m ensured a reliable link to the
satellite data with similar spatial resolution, as we followed the recommendations on ESU. The
RGBNIR Sentinel-2 bands have a spatial resolution of 10 m and the red edge (NIR) and SWIR
bands a spatial resolution of 20 m, and even if we downsampled the bands to 10 m the spectral
information is sustained. More information on datasets and their spatial and temporal resolutions are
provided in supplementary Table S3.
The presented datasets are limited by the regional in-situ observations and expert knowledge
collected mainly in the central Lena Delta. The remoteness of the area and extremely difficult
logistics to conduct research in the second terrace and the outer rims of the delta are major reasons
for these limitations. However, the delta is relatively homogeneous in habitat classes that develop
based on underlying geomorphology and the disturbance regime (annual flooding and permafrost
thaw processes). Only one major habitat class is absent from the well studied central Lena Delta and
only occurs across the second terrace. For a formal evaluation of both habitat classification
products, we defined an independent test dataset within the central Lena Delta. The comparisons
show a relatively high accuracy for the central Lena Delta (94%) and a lower accuracy for the entire
Lena Delta classification (85%). While this decrease in accuracy was expected, due to the large
spatial extent of the Lena Delta, the limitation of independent evaluation restricted to the central
Lena Delta should be noted. Particularly for the smaller patchy habitat types the accuracy is likely
overestimated. For the large-scale patterns and the dominant habitat types, we are confident that the
classification results are reliable and accurate (see also visual evaluation in Figures S7-S9).
In-situ observations (Datasets 1-3) as well as mapping products (Datasets 4-6) represent conditions
and vegetation composition of 2018. The timing of the summer 2018 expedition coincided with a
relatively high number of cloud free Sentinel-2 images necessary for a high quality habitat
classification. Overall, the described datasets are of appropriate quality to serve as a basis for
additional studies and most importantly as a baseline to identify changes in the future.
# 5 Conclusions
The described datasets provide coherent and complementary information of the major habitat
classes in the Lena Delta in Arctic Siberia, the largest delta in the Arctic. Based on extensive
knowledge collected during fieldwork that included habitat-related measurements of plant
composition, biomass, and hyperspectral field measurements we provide a validated and high-
resolution habitat classification map of the delta. In addition, we linked ecologically important
characteristics of disturbances in the delta to habitat classes, providing a baseline for future studies
of Arctic change as well as a foundation for potential upscaling of related processes such as
biodiversity, ecosystem functions, and biochemical dynamics such as greenhouse gas emissions.
With this update of previous land cover and habitat-related mapping products of the Lena Delta we
strive to facilitate and promote future investigations leading to a better understanding of this highly
sensitive arctic delta system.
# Acknowledgements
Field work in the Lena River Delta was conducted in the frame of the Russian-German LENA
Expeditions based at Research Station Samoylov Island. We thank all colleagues and station staff
involved in the organization and logistics for their great support.
# Code/Data availability
Dataset 1: Shevtsova et al., 2021a, https://doi.pangaea.de/10.1594/PANGAEA.935875, Foliage
projective cover of 26 vegetation sites in the central Lena Delta from 2018, is published as Foliage
projective cover for all major taxa estimated as percent, as tab-delimited text files.
Dataset 2: Shevtsova et al., 2021b, https://doi.pangaea.de/10.1594/PANGAEA.935923, Total above-
ground biomass of 25 vegetation sites in the central Lena Delta from 2018, is published as biomass
aboveground dry mass per major taxa, as well as for 'moss and lichen', 'litter' and the remaining minor
taxa (called 'other plants') and the total biomass in the units [g/m2], as tab-delimited text.
Dataset 3: Runge et al., 2022, https://doi.pangaea.de/10.1594/PANGAEA.945982, Hyperspectral field
spectrometry of Arctic vegetation units in the central Lena Delta, is published as an overview of the
plot details and field spectrometer reflectance spectra in the unit [%] of 28 vegetation plots, as tab-
delimited text files.
Dataset 4: Landgraf et al., 2022 a,b,c. The Sentinel-2-derived central Lena Delta land cover (habitat)
classification consists of the following three data publications: i) Landgraf et al. 2022a,
https://doi.pangaea.de/10.1594/PANGAEA.945056: a raster file with assigned land cover classes and an
ESRI polygon shape file containing the 10 training classes representing the different vegetation
compositions, as geotiff file. Both datasets are based on 2018 satellite images and informed by the in-situ
vegetation plots and expert knowledge. Datasets are in Universe Transverse Mercator (UTM) Zone 52
North projection. ii) Landgraf et al. 2022b, https://doi.pangaea.de/10.1594/PANGAEA.945054. This data
set includes training elements representing different vegetation composition in the form of Elementary
Sampling Units ESUs: 69 pseudo ESUs set with expert knowledge from the field and from Lena Delta
expedition field reports. iii) Landgraf et al. 2022c, https://doi.pangaea.de/10.1594/PANGAEA.945055.
This data set includes training elements representing different vegetation composition in the form of
Elementary Sampling Units ESUs: 23 true ESUs representing the LD18 vegetation plots.
Dataset 5: Lisovski et al., 2022, https://doi.pangaea.de/10.1594/PANGAEA.946407. The Lena Delta
Habitat Map (2018, Sentinel-2) contains i) the Lena Delta habitat map (13 classes), ii) the sand probability
map, both as geotiff files in WGS84 geographic projection, iii) the habitat class description as comma
delimited csv table, and iv) the training dataset (n = 4 278 classified pixels) in geographic decimal
coordinates comma delimited csv table. The data collection also contains the Lena Delta Region of
Interest (ROI) ESRI shapefile outlining the Lena Delta including a coastal water buffer.
Dataset 6: Heim and Lisovski, 2023, https://doi.org/10.5281/zenodo.7575691.  The Lena Delta habitat
disturbance regime map is published in the form of two geotiff files (tiles) in WGS84 geographic
projection.
Code developed in Google Earth Engine to derive habitat classes based in the central Lena Delta
classification, as well as R code for figures can be accessed from the following repository: Lisovski,
S. (2024). Code for 'A new habitat map of the Lena Delta in Arctic Siberia based on field and remote
sensing datasets'. V0.1. Zenodo. 10.5281/zenodo.11197641.

# Competing interests


Birgit Heim is a member of the editorial board of ESSD. Otherwise, we declare no competing interests.

# Funding


SL acknowledges funding from the Geo.X Network for Geosciences in Berlin and Brandenburg. This
study was supported by BMBF KoPf (Grant Number 03F0764B), KoPf Synthesis (Grant Number
03F0834B), and AWI base funds. AR was partially funded by ESA GlobPermafrost and an ESA CCI
postdoctoral fellowship. BH acknowledges HGF REKLIM.

# Authors contribution


SL: Conceptual framework, habitat classification, data analysis, writing
AR: Conceptual framework, field work, spectral field data collection, habitat classification, spectral
data processing, data analysis, writing
IS: Field work, biomass and projective cover measurement in vegetation plots, habitat classification
RRO: Habitat classification
NL: habitat classification, spectral data processing
MF: Field work, spectral field data collection
NiL: habitat class definition, field work
AM: Project management, writing
CS: Spectral data processing
AB: Spectral data processing
UH: Conceptual framework, project management
GG: Conceptual framework, project management, habitat classification, writing
BH: Conceptual framework, field work, habitat classification, project management, writing

# Appendix

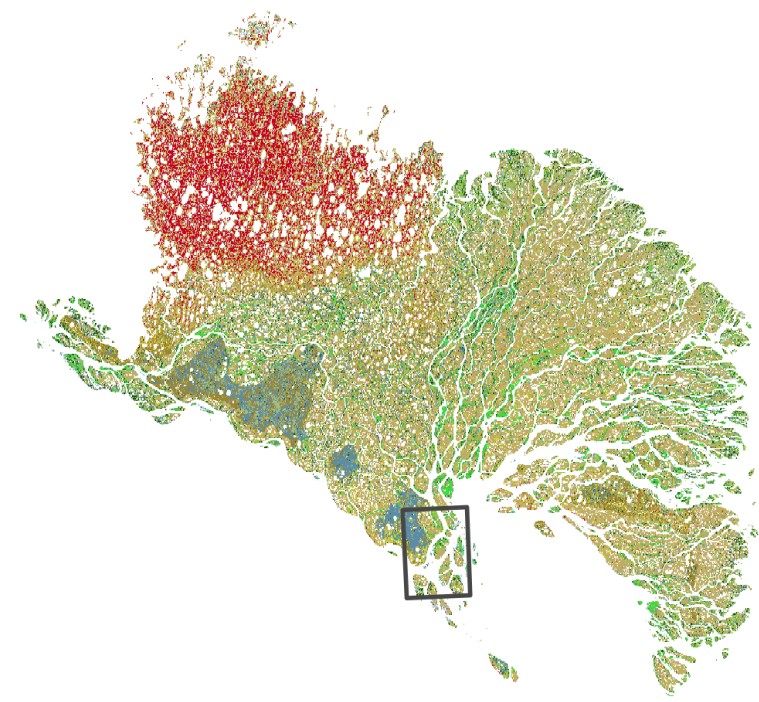

**Figure A1**: *Location of the central Lena Delta habitat classification (Dataset 4) in the Lena Delta (Dataset 5,6).*

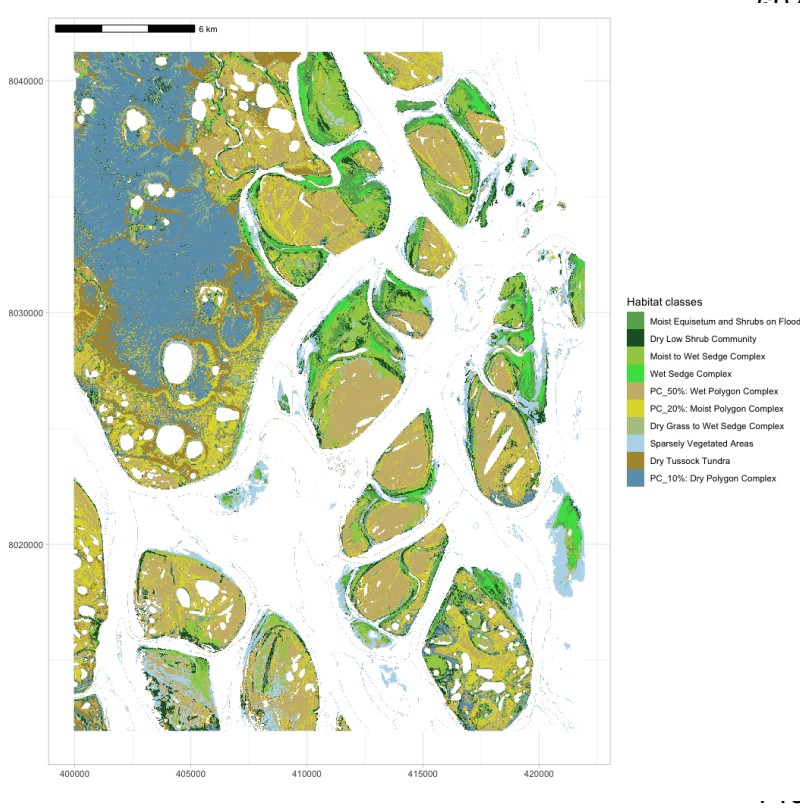

**Figure A2**: *Supervised habitat classification of the central Lena Delta based on a cloud-free Sentinel-2 August 2018 acquisition (Dataset 4). Numbers in legend correspond to the labels in published Dataset 4 (Landgraf et al. 2022).*

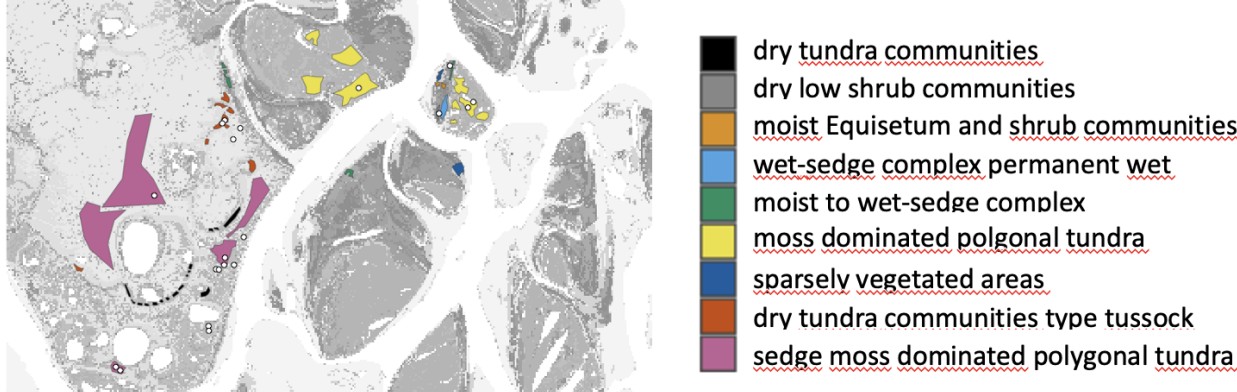

***Figure A3****: The central Lena Delta with 30 x 30 m ESUs (white points, dataset 1) and polygonal shapefiles defined*
*by expert knowledge (published with dataset 4). Together the ESUs and polygonal shapefiles served areas to sample*
*8 626 training pixels for the central Lena Delta landcover/habitat classification (dataset 4, Landgraf et al. 2022a).*

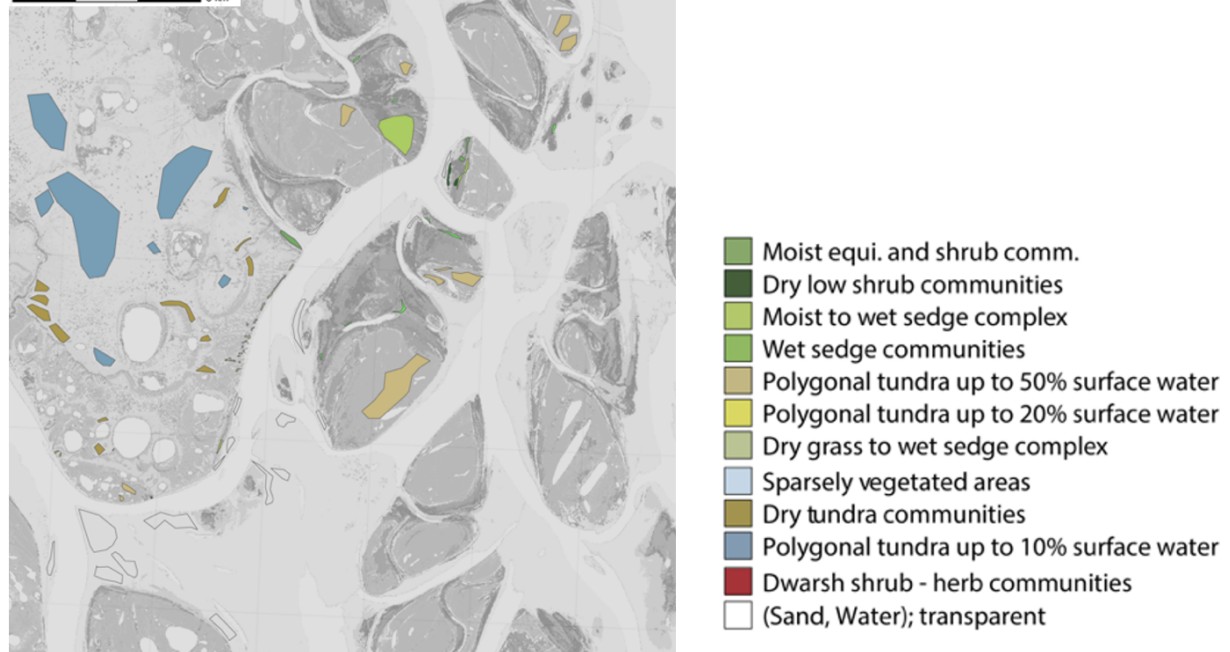

***Figure A4****: Central Lena Delta with the additionally defined polygonal shapefiles as a test dataset for independent*
*evaluation. The polygonal shapefiles were defined using high resolution satellite and drone images, and extensive*
*knowledge from the field (Heim et al. 2025).*

**Table A1**: *Confusion matrix and statistics of the central Lena Delta habitat classification with independently*
*defined polygons (Figure A4, Heim et al. 2025). Statistics are based on 100 random samples per class (1 100*
*samples). Classes refer to 0 = Moist equi. and shrub community, 1 = Dry low shrub community, 2 =Moist wet sedge*
*community, 3 = Wet sedge community, 4 = Polygonal tundra (50%), , 6 =Dry grass and wet sedge complex , 8 =*
*Dry tundra communities, 9 = Polygonal tundra, 10 = Sand.*

| Overall Statistics: | | | | | | | | | |
|---|---|---|---|---|---|---|---|---|---|
| Accuracy | 0.94 | | | | | | | | |
| 95% CI | (0.9223 – 0.9548) | | | | | | | | |
| No Information Rate | 0.1199 | | | | | | | | |
| P-Value [Acc > NIR] | < 2.2e-16 | | | | | | | | |
| Kappa | 0.9325 | | | | | | | | |
| **Class** | **0** | **1** | **2** | **3** | **4** | **6** | **8** | **9** | **10** |
| Sensitivity | 0.8953 | 0.9300 | 0.8774 | 0.8835 | 0.9697 | 0.9278 | 0.9899 | 0.9894 | 1.0000 |
| Specificity | 0.9774 | 0.9987 | 0.9949 | 0.9885 | 0.9975 | 0.9873 | 0.9975 | 0.9911 | 1.0000 |
| Pos Pred | 0.8105 | 0.9894 | 0.9588 | 0.9100 | 0.9796 | 0.9000 | 0.9800 | 0.9300 | 1.0000 |
| Neg Pred | 0.9886 | 0.9911 | 0.9835 | 0.9847 | 0.9962 | 0.9911 | 0.9987 | 0.9987 | 1.0000 |
| Prevalence | 0.8105 | 0.9894 | 0.9588 | 0.9100 | 0.9796 | 0.9000 | 0.9800 | 0.9300 | 1.0000 |
| Detection Rate | 0.8953 | 0.9300 | 0.8774 | 0.8835 | 0.9697 | 0.9278 | 0.9899 | 0.9894 | 1.0000 |
| Detection Prevalence | 0.8508 | 0.9588 | 0.9163 | 0.8966 | 0.9746 | 0.9137 | 0.9849 | 0.9588 | 1.0000 |
| Balanced Accuracy | 0.0973 | 0.1131 | 0.1199 | 0.1165 | 0.1120 | 0.1097 | 0.1120 | 0.1063 | 0.1131 |

**Table A2:** *Confusion matrix and statistics of the entire Lena Delta habitat classification with independently defined*
*polygons (Figure A5). Note, that polygons are from the central Lena Delta only and evaluation statistics are only*
*representative for a small spatial subset of the entire Lena Delta. Statistics are based on 100 random samples per*
*class (1 100 samples). Classes refer to 0 = Moist equisetum and shrub community, 1 = Dry low shrub community, 2*
*=Moist wet sedge community, 3 = Wet sedge community, 4 = Polygonal tundra (50%), 6 =Dry grass and wet sedge*
*complex, 8 = Dry tundra communities, 9 = Polygonal tundra.*

| Overall Statistics: | | | | | | | | |
|---|---|---|---|---|---|---|---|---|
| Accuracy | 0.8431 | | | | | | | |
| 95% CI | (0.8157 – 0.8679) | | | | | | | |
| No Information Rate | 0.1722 | | | | | | | |
| P-Value [Acc > NIR] | < 2.2e-16 | | | | | | | |
| Kappa | 0.8207 | | | | | | | |
| **Class** | **0** | **1** | **2** | **3** | **4** | **6** | **8** | **9** |
| Sensitivity | 0.742 | 0.932 | 0.689 | 0.702 | 0.912 | 0.956 | 1.000 | 0.960 |
| Specificity | 0.957 | 0.961 | 0.989 | 0.980 | 0.999 | 0.951 | 0.991 | 0.996 |
| Pos Pred | 0.688 | 0.708 | 0.930 | 0.870 | 0.989 | 0.650 | 0.939 | 0.970 |
| Neg Pred | 0.967 | 0.993 | 0.939 | 0.946 | 0.987 | 0.996 | 1.000 | 0.994 |
| Prevalence | 0.688 | 0.708 | 0.930 | 0.870 | 0.989 | 0.650 | 0.939 | 0.970 |
| Detection Rate | 0.742 | 0.932 | 0.689 | 0.702 | 0.912 | 0.956 | 1.000 | 0.960 |
| Detection Prevalence | 0.714 | 0.805 | 0.791 | 0.777 | 0.949 | 0.774 | 0.969 | 0.965 |
| Balanced Accuracy | 0.114 | 0.093 | 0.172 | 0.158 | 0.130 | 0.087 | 0.119 | 0.128 |


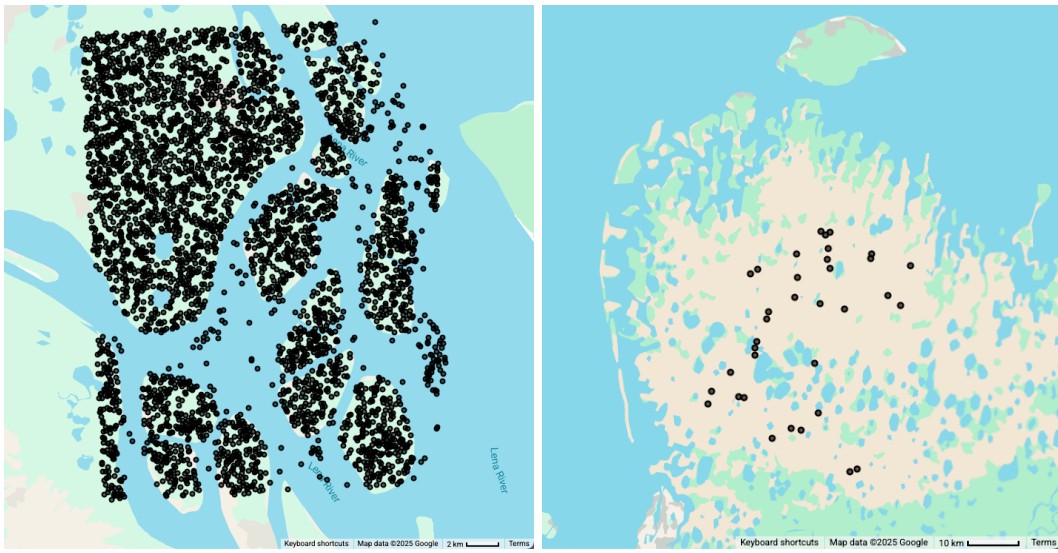

***Figure A5****: Training pixels for the Lena Delta habitat classification (dataset 5). (Left) 7.500 random pixel samples*
*across the habitat classes from the central Lena Delta landcover/habitat map (dataset 4). (Right) 35 pixels (ESUS,*
*Landgraf et al. 2022) selected by expert knowledge for the 'dwarf shrub - herb communities' that are missing in the*
*central Lena Delta.*

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

# Figures

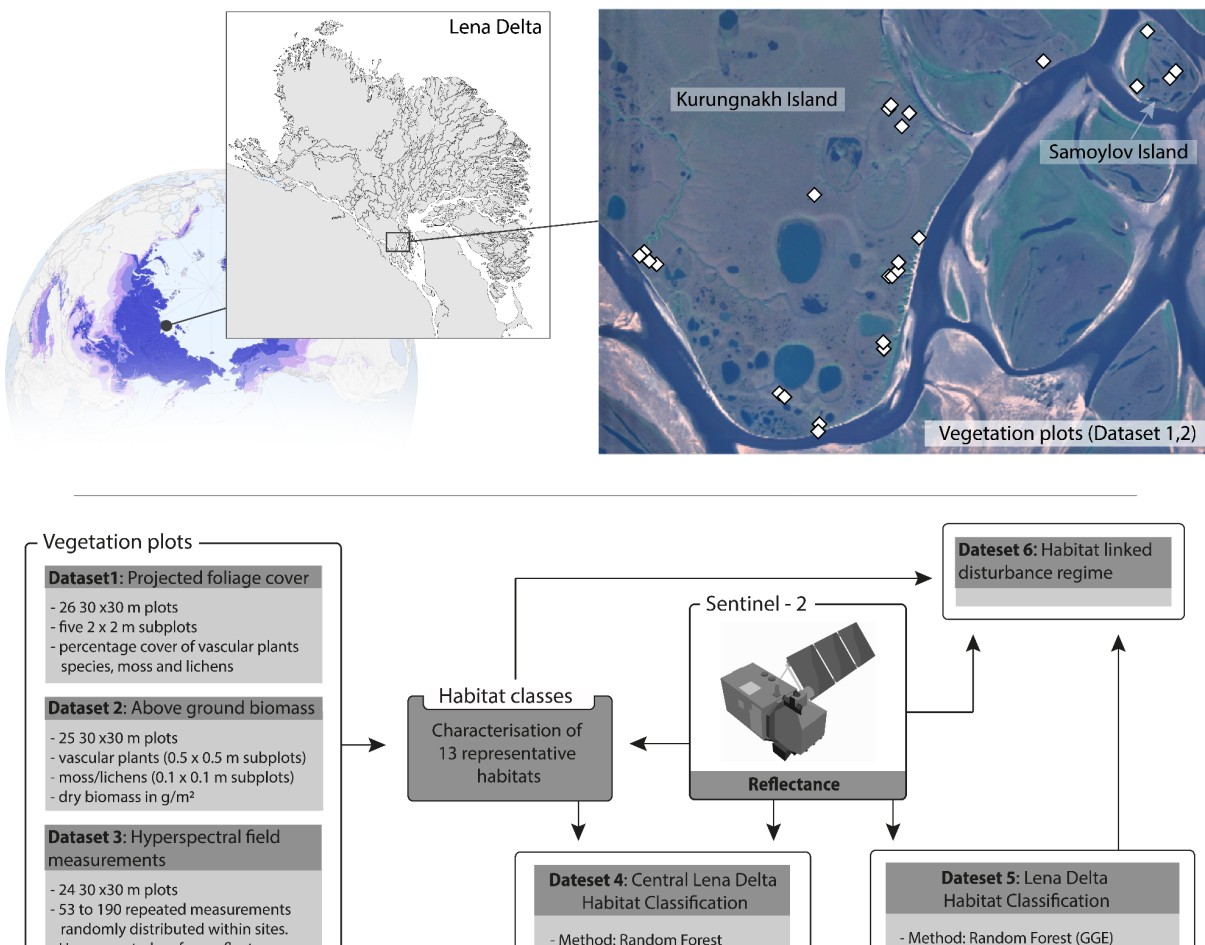

Figure 1: Geographic location of the Lena Delta in the Russian High Arctic (72.91ºN, 126.90ºE)
and a Sentinel-2 RGB image (August 2018, bands 4-3-2) of the central Lena Delta showing the
areas of the 26 vegetation plots where foliage projective cover and above ground biomass was
determined. Panarctic overview map shows permafrost extent (colour scale indicates
permafrost extent from continuous (dark purple) to isolated (light purple) (Obu et al., 2020). The
grey-coloured Lena Delta land map created with Sentinel-1 water mask from Juhls et al. (2021).
Bottom: Dataset characteristics and methodological links between the different datasets.

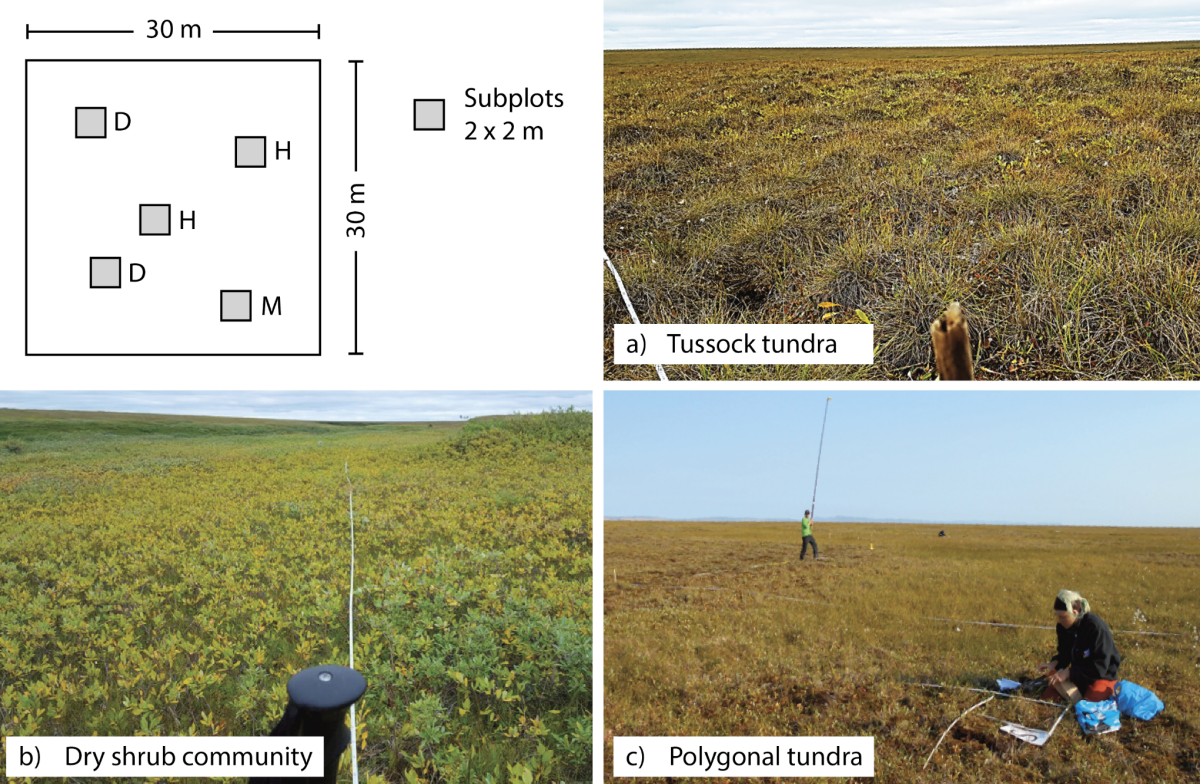


Figure 2. Vegetation plots (30 x 30 m) were established in different vegetation types across the
central Lena Delta. For subplots (2 x 2 m), the projective vegetation cover was recorded and
labeled according to vegetation and moisture properties (H-Type: homogeneous, M-Type:
moist, D-Type: dry). Figures illustrate example plots in a) tussock tundra (VP14), b) dry shrub
communities (VP05), c) polygonal tundra (VP13). Photos: AWI.

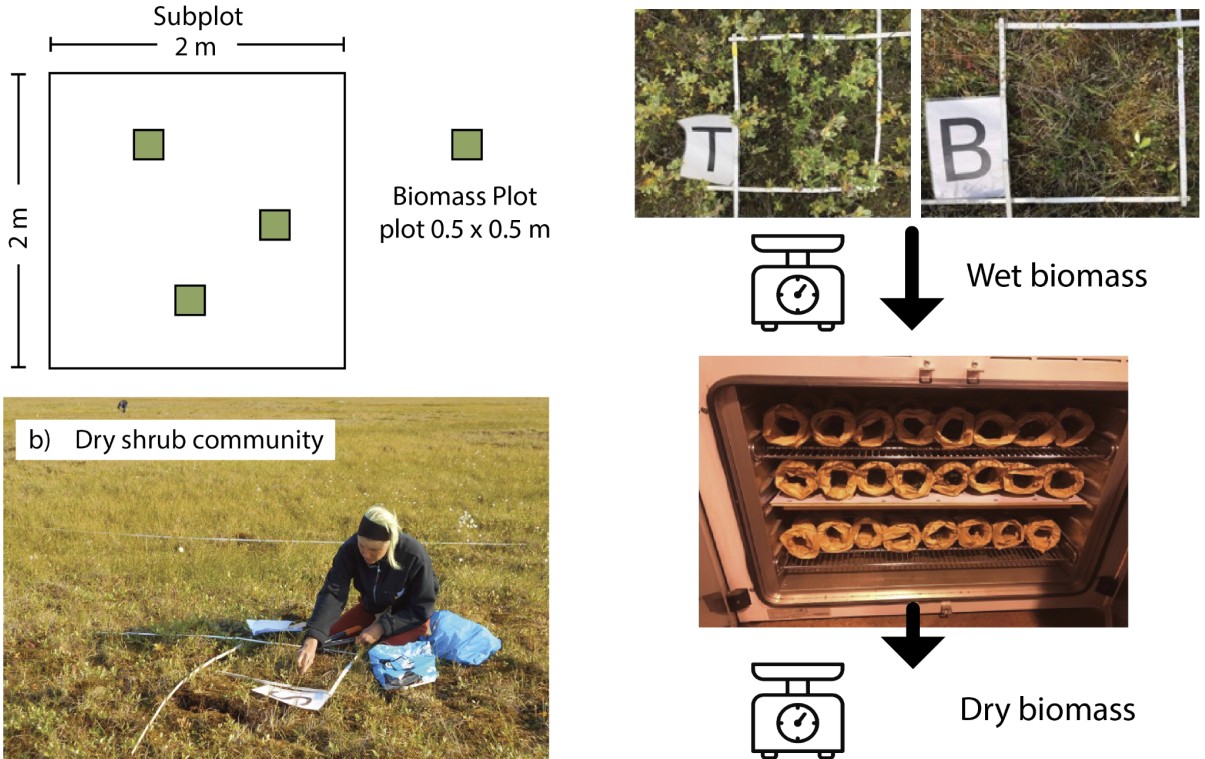


Figure 3. Biomass was sampled in subplots of 0.5 x 0.5 m (and 0.1 x 0.1 m for moss and
lichens) distributed within the 2 x 2 m subplots described in Figure 2. Collected plants were
weighted (wet biomass), dried in an oven and again weighted (dry biomass). Fotos: AWI.

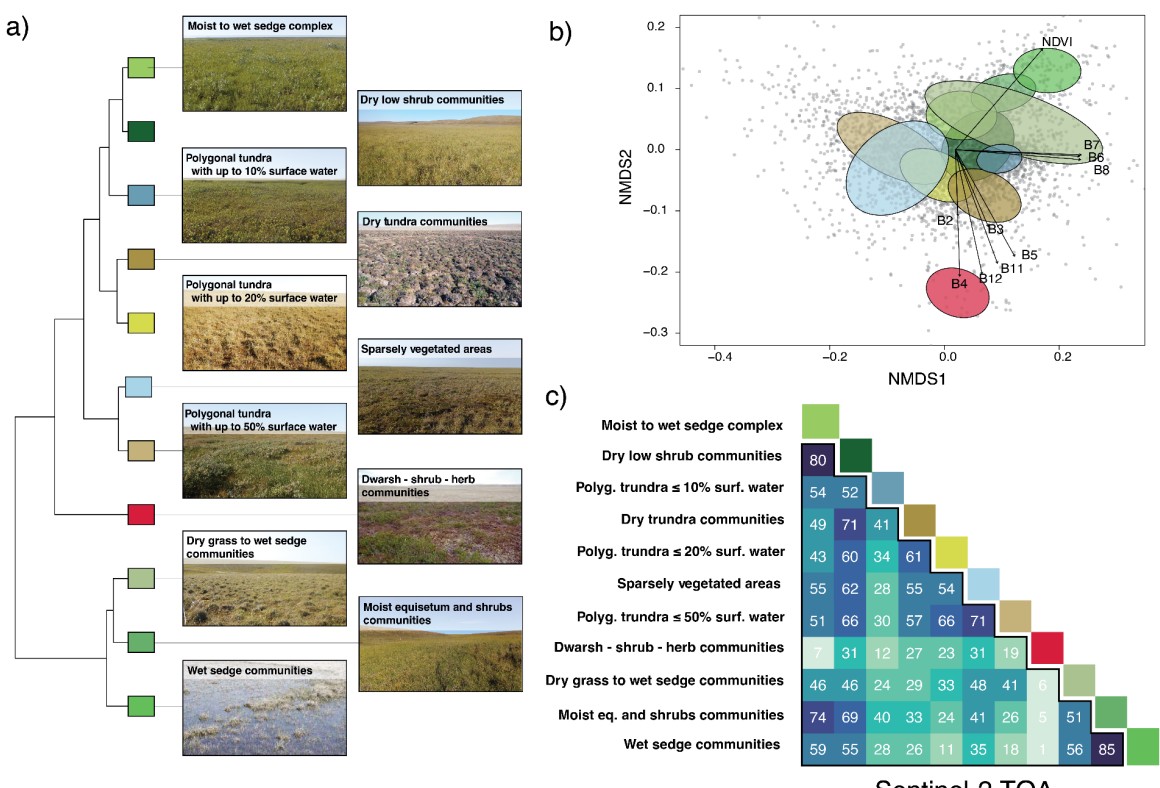


Figure 4: Similarity of habitat classes based on Sentinel-2 spectral reflectance and NDVI values.
The dendrogram in panel a) indicates the multidimensional hierarchical similarity of the classes
based on Sentinel-2 top of atmosphere reflectance (bands 2-8, 10-12, and NDVI). Panel b)
shows the location of the habitat classes within a two-dimensional NMDS space. The arrows
with the Sentinel-2 bands and NDVI indicate the correlation of these variables across the two
axes. The lower matrix of panel c) depicts the calculated percentage overlap of 3,500 pixels
(grey dots in panel b) across the two NMDS axes of panel b).

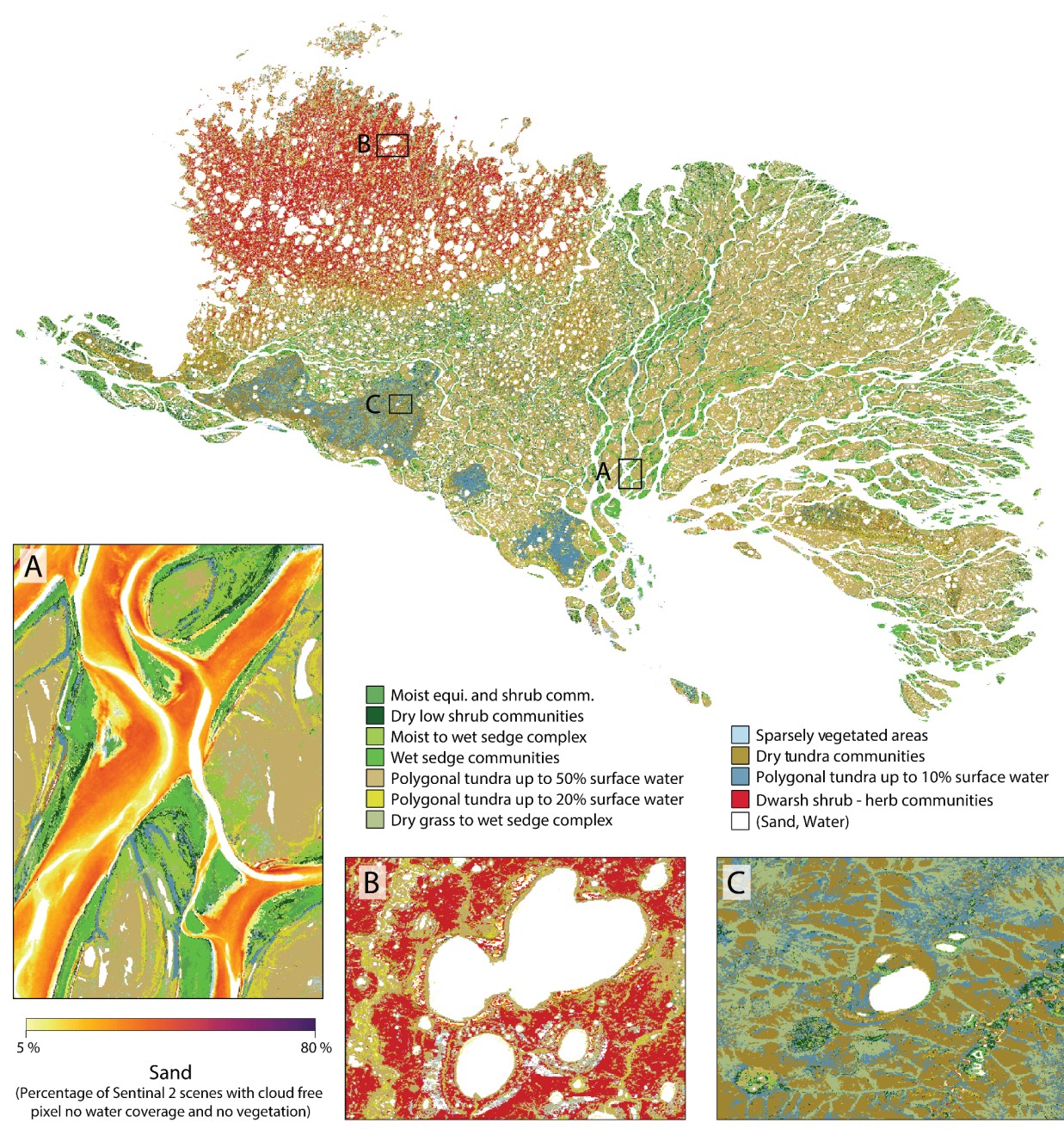

Legend:
- Moist equi. and shrub comm.
- Dry low shrub communities
- Moist to wet sedge complex
- Wet sedge communities
- Polygonal tundra up to 50% surface water
- Polygonal tundra up to 20% surface water
- Dry grass to wet sedge complex
- Sparsely vegetated areas
- Dry tundra communities
- Polygonal tundra up to 10% surface water
- Dwarsh shrub - herb communities
- (Sand, Water)

Sand
(Percentage of Sentinal 2 scenes with cloud free
pixel no water coverage and no vegetation)

5 %        80 %


Figure 5: Lena Delta habitat classes (Dataset 5). The entire Lena Delta on the left with three regional examples, showing (A) the seasonal sand probability and (B, C) regional examples of the habitat classes.

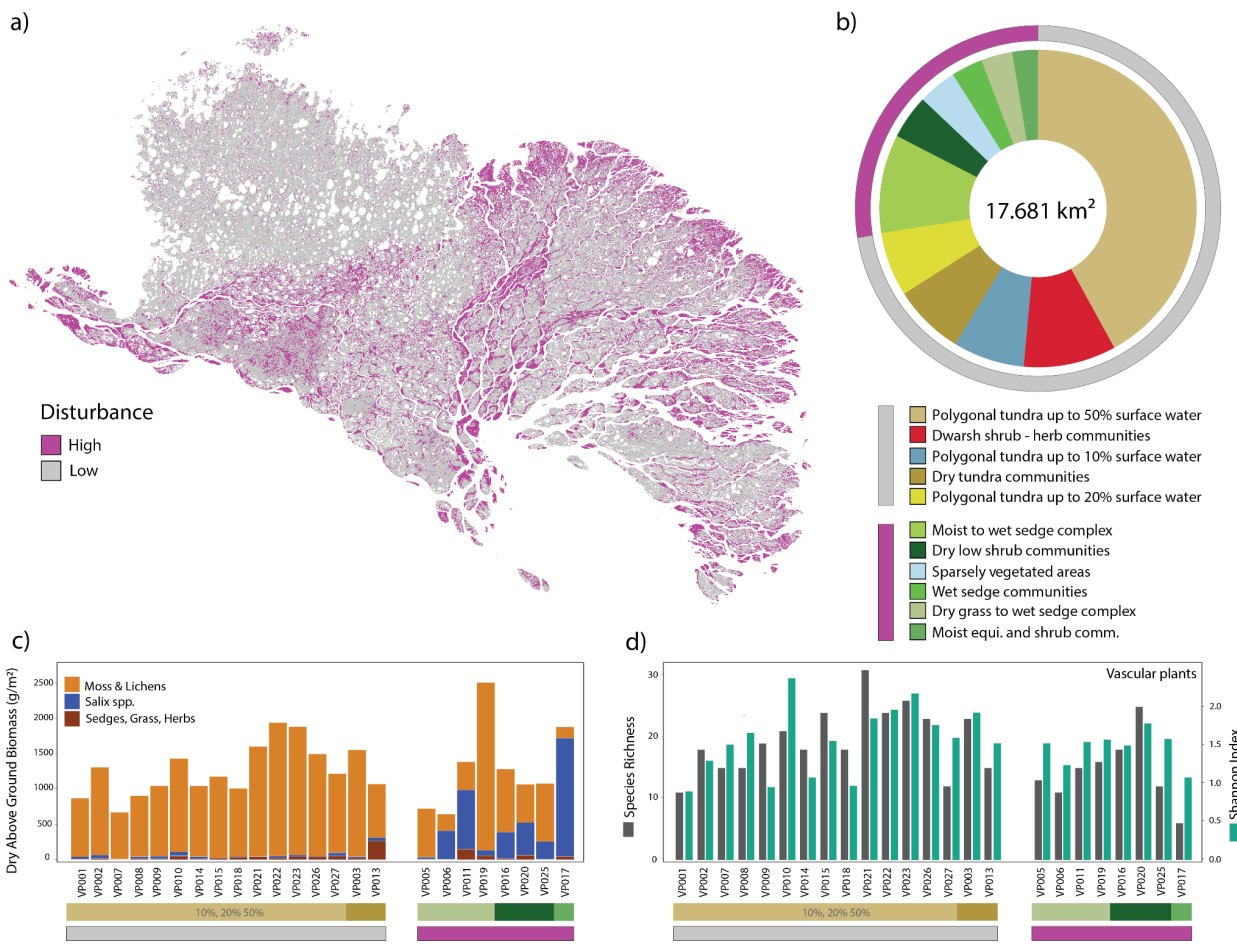


Figure 6: Habitat linked disturbance regimes across the Lena Delta. The map (a) includes all
vegetated areas (excluding water and sand). The pie chart (b) shows the contribution of
vegetated classes across the Lena Delta grouped by high and low disturbance regimes. The
bottom panels show c) the measured dry above ground biomass (Dataset 2) and d) the species
richness and Shannon index (from Dataset 1) of the vegetation plots for different habitat classes
and disturbance regimes.


# Tables


Table 1: Habitat classes, descriptions as well as methods used to characterize the distinct
habitats. In-situ vegetation plot numbers correspond to the vegetation plots of Dataset 1 and 2
(see also Table S1, S2, S3).

| Habitat types | Description | Method |
|---|---|---|
| **Moist *Equisetum* and shrubs** | *Equisetum* and shrub communities form an early-to-middle successional stage growing on the active floodplain. Low moss contribution | In-situ vegetation plot (VP17); extended to representative larger polygon shape files using field knowledge. |
| **Dry shrub communities** | Patch forming shrub communities dominated by dwarf willow (*Salix*) thickets, frequently occurring on dry elevated areas on floodplains and stream floodplains and in topographically sheltered areas below basin and valley rims. Low moss contribution | In-situ vegetation plots (VP04, VP16); extended to representative larger polygon shape files using field knowledge. |
| **Polygonal tundra complex** up to<br>- 10%<br>- 20%<br>- 50% surface water<br><br>(3 distinct classes) | Mature-state plant communities dominated by sedge, moss and herb species. Sparse vascular plant coverage (dwarf willows, dwarf birches) on thick continuous moss cover. Occurring on the plateaus of the ice-rich holocene and pleistocene terraces, and at the bottom of alases. Intersected by intra- and interpolygonal ponds resulting in up to 10%, 20%, 50% surface water contribution. | In-situ vegetation plots (VP01, VP02, VP07, VP08, VP14, VP15, VP18, VP21, VP22, VP23, VP26, VP27); extended to representative larger polygon shape files using field knowledge. The different surface water contributions were defined based on the result from unsupervised classification. |
| **Dry grass to wet sedge communities** | These early-to-middle successional plant communities cover unstable valley slopes and a young drained lake basin, they are mostly composed of sedges and grasses, but also willows (*Salix)* are part of this habitat. | In-situ vegetation plots (VP05, VP06, VP11, VP19, VP20); extended to representative larger polygon shape files using field knowledge. |
| **Dry tundra communities** | The mature-state dry tundra communities represent the zonal tundra type, one subclass is dominated by tussock forming *Eriophorum* and the other by less tussock forming dry-herb communities, dominated by *Dryas*. Occurring on well-drained slopes of valleys and alases, and other well-drained areas on the terraces. High moss contribution | In-situ vegetation plots (VP03, VP13) extended to representative, larger polygon shape files using field knowledge (including 'dry tundra communities type tussock' and 'dry tundra communities'). |
| **Moist to wet sedge communities** | These mid to advanced successional plant communities occur on moist to water-logged soils characteristically mostly in topographic depressions on the floodplains, in valleys and alases. They constitute the rims of the wetland areas on | Polygon shape files derived from high resolution satellite image and ESRI GE with regional expert knowledge. No vegetation plots (too wet). |

| | | |
|---|---|---|
| | the floodplains in more dynamic parts the moss ground cover is missing. | |
| **Wet sedge communities** | These mid to advanced successional plant communities occur at permanently wet sites with stagnant water in the topographic depressions and are typical for wetland areas on the floodplains. In more dynamic parts the moss ground cover is missing. | Polygon shape files derived from high resolution satellite image and ESRI GE with regional expert knowledge. No vegetation plots (too wet). |
| **Sparsely vegetated areas** | These early successional plant communities are characterized by low vegetation establishment and coverage. No to low moss contribution | Defined based on the result from unsupervised classification, polygon shape files. No vegetation plots. |
| **Barren/Sand** | Representing the wide-open sand flats of the floodplain and barren ground on valley slopes or along cliffs. In a few cases, this class represents vegetation-free bedrock outcrops. | Threshold using high reflectance in S2-band 2 blue. |
| **Water** | Represents all surface water bodies in the delta: the Lena River with river branches, streams, lakes and large ponds. | Threshold using low reflectance in S2-band 8 NIR. |


Table 2: Habitat class and description of disturbance regimes and the component stand
structure in form of contributions of vascular plants, and moss to total biomass. * (Driscoll and
Hauer, 2019; Stanford et al., 2005)*, ** (Lorang and Hauer, 2006).

| Habitat class | Disturbance regime | Stand structure |
|---|---|---|
| Moist *Equisetum* and shrubs | **High; regular (annually), predicted**<br>- spring floodings,<br>- shifting habitat *<br>- advanced-stage regeneration ** | high vascular plant growth, low abundance of moss & lichens. |
| Dry shrub communities | **High; mixed disturbance types**:<br>-regular spring floodings<br>-rapid thaw processes (permafrost degradation)<br>- shifting habitat<br>- advanced-stage regeneration | high vascular plant growth, low abundance of moss. |
| Polygonal tundra complex | **Low; mixed disturbance types**<br>- low for most of the habitat, except for actively eroding shores of ponds and channels<br>- mature-state plant community | low vascular plant growth,<br>high abundance of moss. |
| Dry grass to wet sedge communities | **High; mixed disturbance types**:<br>- regular spring floodings<br>- rapid thaw processes (permafrost degradation)<br>- shifting habitat<br>- advanced-stage regeneration | high vascular plant biomass, low abundance of moss. |
| Dry tundra communities | **Low; mixed disturbance types**<br>- low for most of the habitat<br>- mature-state plant community | low vascular plant biomass<br>high abundance of moss. |
| Moist to wet sedge communities | **High; mixed disturbance types**:<br>- regular spring floodings<br>- rapid thaw processes (permafrost degradation)<br>- shifting habitat<br>- mid to advanced-stage regeneration | high vascular plant biomass Almost impossible to measure in-situ biomass (wet conditions and difficult access). |
| Wet sedge communities | **High; mixed disturbance types**:<br>- regular spring floodings<br>- rapid thaw processes (permafrost degradation)<br>- shifting habitat<br>- mid to advanced-stage regeneration | high vascular plant biomass. Almost impossible to measure in-situ biomass (wet conditions and difficult access). |
| Dwarf shrub herb communities | **Low; mixed disturbance types**<br>- low for most of the habitat<br>- mature-state plant community | low vascular plant biomass, high abundance of moss. |
| Sparsely vegetated areas | **Very high; mixed disturbance types**<br>- regular spring floodings | lowest vascular plant biomass, no moss. |

| | | |
|---|---|---|
| | - rapid thaw processes (permafrost degradation)<br>- shifting habitat<br>- early-stage regeneration | |
| **Sand banks/barren** | **Very high: mixed disturbance types**<br>- regular spring floodings<br>- rapid thaw processes (permafrost degradation)<br>- shifting habitat<br>- no regeneration | Barren, constant shifting of sediments and movement of soils. |
