# Peer review of "A new habitat map of the Lena Delta in Arctic Siberia"

_Earth System Science Data, 2023_

## Author Comment (AC1)

**Reviewer #2:**

General Comments:

The manuscript provides valuable insights into the ecological changes within the Lena Delta, leveraging satellite imagery and field data to map vegetation and habitat distributions. This work is timely and contributes significantly to our understanding of Arctic ecosystems' responses to climate change. However, several areas require further detail and clarification to enhance the manuscript's impact and utility for future research.

Major Comments:

Data Source and Acquisition Transparency:

Suggestion: Provide detailed information on data sources, including satellite data acquisition dates, sensor types, and ground truth data origins. This detail will help verify the reliability and applicability of the data used.

- Thank you, we included a table in the supplements that highlights the different datasets, their inputs and sources. In addition, we provide more details in the data availability chapter. The revised figure 1 should also provide a better description and support for understanding the individual datasets as well as their differences.

Methodology on Multi-source Data Fusion:

Suggestion: If the study integrates multiple data sources (e.g., different satellite sensors, ground measurements), describe the fusion methods, including how resolution, coverage time, and accuracy differences were addressed.

- Spatial Interpolation and Scaling Methods:

Suggestion: Detail the spatial interpolation or scaling methods used, their justification, and potential impacts on dataset accuracy and reliability.

- We combined the two review comments as they refer to the methods. Sorry about the confusion, we realized that the former version of Figure 1 could be misleading, making it seem that we applied technical data fusion. In this study, we did not apply a technical data fusion method, specifically not a pixel-level multi-scale data fusion. We used the field data to decide which habitat classes could be assessed and how to characterize them (e.g. table 2 in the manuscript). We named this process in the former Figure 1 'habitat characterisation' with arrows leading from the boxes of the 3 field data products to the box of 'habitat characterisation'. New: we now deleted the two arrows that are the cause for confusion (one arrow leading directly from the field

data to the central Lena Delta classification (Landgraf et al. 2022), one arrow leading directly from the field data to the disturbance map (Heim and Lisovski 2023).

Revision: i) updated figure 1; ii) 2 new technical figures in appendix visualizing data sources and steps to produce a) the central Lena Delta classification and b) the Lena Delta habitat map  and iii) new table S3 in appendix with detailed overview on input data and characteristics.

Furthermore, we added a new subsection '4.4 Classification accuracy and representativeness' to also address other comments and to further discuss the use of field work and satellite data (LL560-572).

Temporal and Spatial Scale:

Temporal-Spatial Coverage and Representativeness:

Suggestion: Discuss whether the dataset's temporal-spatial coverage adequately represents the Lena Delta's seasonal and multi-year variations and its potential impact on ecosystem change analysis.

- We included an entire section in the discussion dealing with temporal and spatial accuracy and representativeness of our datasets.
-

Choice of Temporal-Spatial Resolution:

Suggestion: Justify the chosen temporal-spatial resolution, including how the balance between data processing capabilities and analysis needs was achieved and the potential impact of different resolutions on result interpretation.

- These are very good points. We discuss this in the text LL513-524 and LL 573-584

- The land cover map from Schneider et al. (2009) is based on two cloud-free Landsat images from June/July 2000 and 2001, the ESA CCI land cover 2018 map is based on summer images as well. Hence, the images used for this habitat classification were acquired at a similar time as for the ESA CCI product and we do not expect differences based on changes on the ground due to this temporal concurrence. In the almost 20-year difference between Schneider et al. (2009) and this habitat map we do expect changes in vegetation composition. Overall, it is challenging to obtain sufficient cloud-free images during the summer months to fully cover the entire Lena

Delta for a classification project and to depict a specific phenological state. Therefore, we created a Sentinel-2 composite mosaic based on the maximum NDVI value per pixel from June to September. With this we ensure to have the peak vegetation and phenology season represented as input for the habitat classification as much as possible and increase comparability to other classification studies despite a temporal mismatch.

- The field data was acquired during a field trip in July-August 2018, primarily focusing on 30 m x 30 m homogeneous vegetation and land cover plots. Additionally, we relied on Sentinel-2 images for the different classifications that were also acquired in summer 2018, covering the same period as the field trip, and have a spatial resolution of 20 m. The temporal overlap of the field work and the satellite image acquisitions ensures consistency across the different datasets and represents a close relationship between datasets and products obtained in the field (dataset 1, 2 and 3) and the results derived from the satellite images that use the field data as input. As Sentinel-2 images have a small geolocation error, we could link our field plot locations directly with the satellite images. Furthermore, the sampling and measurement design of the plots with 30 m x 30 m ensured a reliable link to the satellite data with similar spatial resolution, as we followed the recommendations on ESU. The RGB NIR Sentinel-2 bands have a spatial resolution of 10 m and the red edge (NIR) and SWIR bands a spatial resolution of 20 m, and even if we downsampled the bands to 10 m the spectral information is sustained.

Accuracy and Uncertainty:

Accuracy Assessment:

Suggestion: Include additional accuracy assessment metrics beyond classification accuracy, such as user accuracy, producer's accuracy, and Kappa coefficient, to comprehensively evaluate the dataset's quality.

- We have revised our approach to assess classification accuracy. Several reasons lead to the fact that we are not able to perform independent classical cross-validation and quantitative assessments. We have, however, discussed the accuracy etc. in more detail and added an entire section in the discussion.

Uncertainty Analysis:

Suggestion: Conduct a quantitative analysis of uncertainties in the dataset, including those arising from data sources, classification methods, and choice of temporal-spatial resolutions. Discuss the potential impact of these uncertainties on ecosystem analysis and interpretation.

- Concerning the question of temporal-spatial resolution, please see the answer to your previous comment.

Minor Comments:

Language and Expression: Ensure consistency in terminology throughout the manuscript. For instance, clarify the use of "habitat types" vs. "vegetation types" and maintain consistent use throughout.

- Thanks, you are right, we were not quite consistent within our manuscript. We made changes throughout the document for consistency. However, other studies often performed land cover classifications and called them accordingly, hence, we used this term when comparing to our habitat classification.

Figures and Tables: Enhance the readability of figures by adjusting label sizes and including legends directly on figures for clarity. Ensure tables detailing methodologies are clear and abbreviations are defined.

- Thank you. We adapted the figures and ensured increased readability.

Supplementary Information: Consider adding supplementary material detailing the technical specifications of satellite images, field equipment, and data analysis algorithms.

- Thanks, we created the above described overview table for the supplementary material.

Conclusion:

This study presents important findings on the Lena Delta's ecological dynamics. Addressing the above suggestions will strengthen the manuscript, making it a valuable resource for the scientific community interested in Arctic ecosystems and climate change impacts.

- Thank you very much for your constructive review and feedback.

---

## Author Comment (AC2)

**Reviewer #1:**

Arctic wetland ecosystems are vulnerable to climate change, but there are still not a few datasets in the region to study the feedback between these ecosystems and climate change. This study provides a few useful datasets about the vegetation in the Lena Delta, which is a unique region. Particularly, the detailed habitat type map will potentially be the most valuable product provided by this study, which, unfortunately, raises me a few concerns about the quality.

The download links for these datasets provide data in tab-delimited text, I'd suggest also provide in other more readable format, such as excel-readable csv, and particularly for maps, in the format of geotiff or other commonly used image format, and a preview in map form (currently, the preview is not working).

- The datasets were uploaded based on the defined and required formats of the Data Publisher for Earth & Environmental Sciences PANGAEA. However, we understand that the format might not always be intuitive and also the downloading options and processes are sometimes confusing. We have therefore expanded the data availability section in the main text to better describe the file formats (e.g., tab-delimited text files for Shevtsova et al., 2021a, Shevtsova et al., 2021b, Runge et al., 2022, and the training data in Landgraf et al., 2022b, geoTIFF and shapefiles for the other datasets). Please see the revised text in Line 601ff.

  In addition, we added an example (with screenshots) on how to download the data in the supplementary material.

Line 155, 162-163. it is confusing here about homogeneous vegetation type. It is homogeneous at what level? It is supposed to only have one "habitat class" as defined in dataset 4 or what? Also, it is stated that in line 155 all 30×30m plots are homogeneous vegetation types, then later in lines 162-163, there are different methods applied to the plot if vegetation cover was homogenous and if vegetation cover is more diverse.

Dataset 1: The vegetation cover was recorded or measured at the center of each 30×30m plot with a ring of 50cm, and then scaled up to the whole plot. How is this done? And how floristic composition played a role in this process. Would the 30×30m plot include more vegetation species than the center 50cm-radius subplot?

Dataset 2: Again, I think more information about the scaling to the 30×30m plot is needed, and why it is reliable.

Figure 2: Can you label the length of edges of these squares, so we may immediately understand which one is the main 30×30 m plot, and which are smaller-sized subplots? I could not tell which were 2×2 m plots and, which were 0.5×0.5 m plots.

-   We understand the confusion and revised the text as well as the figure captions. Specifically, we provide a more detailed description of Dataset 1 and 2, and provide a revised Figure 2 according to your comments and to better highlight and describe what is meant by 'homogeneous' vegetation types. An additional Figure 3 was produced for Dataset 2, showing more examples of the different plot types and methods applied in the field (as well as the spatial scales for biomass sampling).

[Figure]

Figure 2. Vegetation plots (30 x 30 m) were established in different vegetation types across the central Lena Delta. For subplots (2 x 2 m), the projective vegetation cover was recorded and labelled according to vegetation and soil properties (H-Type: homogeneous, M-Type: moist, D-Type: dry). Figures illustrate example plots in a) tussock tundra (VP14), b) dry shrub communities (VP05), c) polygonal tundra (VP13). Photos: AWI.

[Figure]

Figure 3. Biomass was sampled in subplots of 0.5 x 0.5 m (and 0.1 x 0.1 m for moss and lichens) distributed within the 2 x 2 m subplots described in Figure 2. Pictures illustrate biomass sampling in b) polygonal tundra c) high shrub communities d) shrub and sedge communities e) dry shrub communities. f) Collected plants were weighted (wet biomass), dried in an oven and again weighted (dry biomass). Fotos: AWI.

Dataset 3: Does each 30×30m plot only include one homogeneous land cover?  Are these the same plots as datasets 1 & 2?

- Yes. With the revision the relationships are better described in the text. In a nutshell, the 30x30m plots only include homogeneous (e.g., dry tundra) and quasi-homogeneous (e.g. polygonal tundra) land cover types. We collected 28 hyperspectral field measurements, 15 of them within the same vegetation plots described in Dataset 1 and 2 (please see supplementary table S2 for an overview). Furthermore, we revisited plots later in August to capture senescence. In addition, we also conducted 10 field-spectroscopy measurements on non-vegetated areas such as sandy parts of the floodplain.

3.4.1: what is a S-2 based supervised classification?

3.4.2: what is S-205 2A Level 2A image? This section, without pre-information of the sensors, makes it hard to understand the different types/levels of images.

3.4.2: all spectral bands were resampled to 10m pixel resolution bands. Which bands are 10-m resolution, and what spectral bands does the sensor have?

- We added additional information on the Sentinel-2 satellite dataset in the text (L105 and copy pasted below), including information which bands have a 10 m and which 20 m pixel resolution. We also removed the too detailed term S-2 Level 2A (referring to the atmospherically corrected Bottom of Atmosphere (BOA) S-2 product level).

  "The availability of Sentinel-2 (S-2) Multispectral Instrument (MSI) data from two orbiting satellite missions since 2016 and 2017 provide high quality multispectral satellite data with a spatial resolution in the Visible and Near Infrared wavelength region of up to 10 m, and of 20 m in the red edge and the Short-wave infrared wavelength regions (Drusch et al., 2012, ESA 2015)."

3..4.3: what is the distribution of the 8,626 training pixels? Are they scattered in the classification area domain? If they are clustered or formed from polygons with the same class, the efficient number of training pixels will be much reduced. A map of the training/validation samples can help understand the situation.

- We understand the concern and the problem of autocorrelation with the method that was applied for the central Lena Delta classification. Not as a justification but to provide background, the already published dataset was developed as part of a bachelor thesis (Nele Landgraf). In the thesis, NL and BH tried to tune a classification model for a region where both (and also additional authors) have very good on-ground knowledge of vegetation types, communities and structures. With the final choice of training points the classification produced a highly accurate map separating the different habitats that are known from fieldwork and remote sensing images. We are aware of the methodological issues, however, given the qualitative assessment of accuracy based on expert knowledge, we considered this product valuable and useful as a training dataset for the entire Lena Delta (for which expert knowledge is very sparse and vegetation monitoring is lacking). In our revision, we added a map showing the training pixels and described the method (appendix), as well as a description of the qualitative nature of the evaluation procedure in the dataset discussion (L275) and a new specific section (L564).

What is ESUS?

- ESUs are Elementary Sampling Units that serve as spatial training and validation units representative for the land surface for quantitative and qualitative remote sensing operations. This process is in accordance with the Committee on Earth Observing Satellites Working Group on Calibration and Validation (Duncanson et al., 2021). Description and reference can be found in the text (L193).

Line 224:  A detailed user's and producer's accuracy report as well as the overall classification accuracy is needed here to justify a good classification. The validation samples should be "independently" distributed from the training sample: i.e., they should not come from the same plot/polygon, because of the well-known auto-correlation problem.

Line 229: it seems you only have 26+69 training locations. Which is quite a small training sample. As I mentioned, counting the number of pixels in a continuously distributed polygon is misleading.

- For the training, S-2 pixels from the 30 x 30 m ESUs representing the 26 vegetation plots (Dataset 1, Shevtsova et al. 2021a), and additional training polygons defined by expert knowledge were used and are all published in the Landgraf et al 2022a data collection. To understand the class and feature performance we applied a cross-validation. As you correctly point out, the overall quantitative class accuracy of 96.78% is this high due to the autocorrelation of neighbouring pixels in the 9-pixel size ESUs (3 x 3 pixel size) and neighbouring pixels within the training polygons. The main quality assessment is not considered by the quantitative assessment but by the qualitative assessments that we kept during the process of the classification, interacting with colleagues who had been in the field in the Lena Delta for many years and are in the list of co-authors. The chosen classifier was able to distinguish all relevant classes and was even able to identify known patchy habitat spots. Therefore, we consider the central Lena Delta classification as reliable.

  See new Appendix.

3.5.2: how large is the area, e.g. at least how many tiles of sentinel-2 images are required to cover the whole study area?

- We added both information. The Lena Delta covers an area of 29873.7 $km^2$ and 15 S2 tiles overlap with the extent.

3.5.3: It is good you have 6,500 random points now for training the classifier, but still bad that they all come from the central Lena Delta. Some evidence or proof is needed to show that "reflectance" from other areas is like those in the central data when they are the same class, regarding the complex "Same Object, Different Spectrum" and "Different Objects, Same Spectrum" problem. What is the relative size of the central Lena Delta compared to the whole Lena Delta?

Using the training data from just the central Lena Delta to train a classifier to classify a large area is acceptable, but validation the result still using the location from central Lena Delta is not acceptable at all. This is because the accuracy of the classification of a class is dependent on its dominance, which varies by region. The training samples should be evenly

scattered out to the whole classification area domain. A map of training/validation points is needed.

- We understand and fully agree that this would be the optimal procedure. However, there are simply no validation points from outside the central Lena Delta. Large parts are not accessible (except by helicopter) and have never been visited by humans. Therefore, the classification method we applied is, to our knowledge, the best way to derive a good quality habitat classification for the entire Delta. We added a discussion on how that lacking in-situ data from other parts of the delta is obviously a caveat of the classification and that quantitative assessments are therefore difficult and in this case largely impossible. We also added the relative area of the training dataset (central Lena Delta) in relation to the entire Delta (Central Lena Delta 644.9 km2 (Vegetation cover 55.2%), Lena Delta 29873.7 km2 (Vegetation cover 58.7%); Vegetated area of the central Lena Delta represented 2% of the vegetated area of the entire Lena Delta).

Line 294, can you provide more details about how the upscaling is conducted?

- Details were added and linked to Figure 5 showing the disturbance map (Dataset 6).

Line 299: here you have it named habitat map, but in the data link you call it land use land cover map. I understand they are the same thing but need to be consistent.

- Lisovski et al., 2022 is in the process to be renamed on PANGEA to have consistent naming (Habitat Map).

Line 306, it is not clear where/what are the hierarchical level 1 in Figure 3a.

- The hierarchy refers to the dendrogram in Figure 3 (now 4) a. Level one is the baseline and each split in the tree is another hierarchical level. We added an explanation in the text.

Figure 3: Why do you use surface reflectance for the classification for producing datase4, but then top of atmosphere reflectance for dataset5? It seems to me it is hard to spectrally differentiate the different habitat classes looking at (b), makes me more suspicious about the overall high accuracy of the classification: I do not think the validation scheme is reliable.

- We used TOA images since at the time of the analysis the northern S2 tiles were not yet processed to BOA images. While BOA is generally better, TOA has been proven to be of sufficient quality to derive high accuracy land cover products.

Figure 3: what are B and C inlets, they're not sand probabilities.

- We think you are referring to figure 4 (now figure 5). Indeed, we labelled inlet B and C incorrectly. These do not show the sand probability but regional examples of the Lena Delta habitat map. We corrected the figure caption. Thanks.

*Figure 5*: *Lena Delta habitat classes (Dataset 5). The entire Lena Delta on the left with three regional examples: A showing the seasonal sand probability and B and C regional examples of the habitat classes.*

---

## Author Response (AR2)

Response to Reviewer #2

Accepted as is.

Reply: We thank reviewer #2 for the positive feedback.

Response to Reviewer #1

Thanks to the authors' revision, the manuscripts have been significantly improved and clarified. However, I still have some concern about the habitat classification of the central Lena delta, and big concern to the whole Lena Delta Habitat classification.

For the central delta habitat classification, the training/validation is based on pixels distributed in the 26 ESUs and additional polygons defined by the expert knowledge. Here: how many additional polygons have been defined, based on what expert knowledge or tools/approaches.

Reply: In the first revision, we added Figure A3 showing the ESUs and the defined polygons.

[Figure]

**Figure A3**: Subset of the central Lena Delta with 30 x 30 m ESUs (white points, dataset 1) and polygons defined by expert knowledge (published with dataset 4). Together the ESUs and polygons served areas to sample 8 626 training pixels for the central Lena Delta landcover/habitat classification (dataset 4, Landgraf et al. 2022a).

We added more information on expert knowledge and methods (L271-276).

Also, if the habitat class can be determined based on this expert knowledge, why not determine extra polygons (or pixels) for independent validation? Or alternatively, if you must split the exiting ESU and expert-identify polygons for training/validation, why not sample by polygons instead of pixels to eliminate the auto-correlation problem. Lastly, I still think a class-based

accuracy matrix is needed. An overall accuracy metric is insufficient to suggest good classification from remote sensing images.

Reply: This is what we have done. We have now added a class-based accuracy table (Table A1).

For the habitat classification of the whole Lena Delta, while I understand it is difficult to validate, this could not be the reason without independent validation. Something must be done. The authors mentioned expert knowledge here again, but still vague. Results were carefully checked with expert knowledge, but what is the result of this check, and how it is checked. Is it possible for experts to interpret manually from very high-resolution satellite images, or aerial photograph, or drone flies at some sites for accuracy assessment? I mean, there should be some quantitative validation metric beyond the central Lena Delta, especially when the central Lena Delta only represented 2% of the vegetated area of the whole Delta. Again, I would not accept an overall high accuracy when there is multiple classes.
I would not recommend publishing the datasets or paper if these critical concerns are not well solved.

Reply: We partly understand the frustration. We were fortunate that only one major habitat class, that covers large areas on the second terrace, was missing in the central lena delta. As you can see in the new supplementary figure S7, it is rather straightforward to select suitable training pixels. Creating additional evaluation would work well for such habitat classes that are less patchy and cover large areas, but even with our expert knowledge from years of work in the central delta, we would not dare to define pixels for the smaller patchier habitat types outside the area we know. However, we have included the confusion matrix results of the classifier for the training dataset (Table A1). We have also added three comparison figures in the supplement showing Sentinel-2 RGB and the results of the classification. We sincerely hope that you agree that within the terraces, the habitat structures are largely repetitive (driven by geomorphological features) and that the classification very precisely picks up these repetitive structures and differences between the terraces (new Figures S6-S8). We agree that a lack of formal and more spatially complete evaluation limits the trust in the habitat classifications, and more so in the pathy smaller habitat types rather than the larger distributed types. Besides the inclusion of the class-based accuracy, we added a few sentences regarding the limitations due to the lack of evaluation datasets across the Lena Delta (L586-590).

Minor: Spell S-2 as Sentiel-2

Reply: Done

Unaddressed comments from last review:
Dataset 1: The vegetation cover was recorded or measured at the center of each 30×30m plot with a ring of 50cm, and then scaled up to the whole plot. How is this done? And how floristic composition played a role in this process. Would the 30×30m plot include more vegetation species than the center 50cm-radius subplot?

Reply: (L195) we defined a 30 x 30 m square plot with a homogeneous or repetitive vegetation composition that was also representative of the wider land surface serving as an Elementary Sampling Unit (ESU).

We specifically selected homogeneous vegetation plots with the aim to enable upscaling.

Dataset 2: Again, I think more information about the scaling to the 30×30m plot is needed, and why it is reliable.

Reply: See reply above.

---

## Author Response (AR3)

Response to Reviewer #2

In my view, after multiple rounds of revisions, the author has addressed all my concerns, significantly enhancing the quality of the paper. It now meets the publication standards, and I recommend accepting it for publication pending the journal's technical review.

Reply: We thank reviewer #2 for the positive feedback.

Response to Reviewer #1

I'm sorry for being so strict and may make the authors frustrate, but I hope this ensures the publication of high-quality data.
I can now accept the situation, given the challenges in obtaining independent training labels for the entire terrace. Although there is no independent accuracy assessment, a thorough discussion of the issue, including the potential caveats and limitations of using the product, would be necessary.

Reply: we agree that an independent evaluation is necessary and we have now tried to accommodate for that by defining an additional independent set of polygons. We used these polygons and another independent dataset (50 sampling points) with defined habitat types, to evaluate the accuracy of both, the central and the entire Lena Delta. As expected the accuracy is lower using the independent data points (~90% for the central Lena Delta and ~80% for the entire Lena Delta classification). We agree that this is a more realistic quantification. For the entire Lena Delta and notably for the smaller habitat types this might still be an overestimation. We discuss this in section  "4.4 Classification accuracy and representativeness".

However, I would still being strict on this:
"Also, if the habitat class can be determined based on this expert knowledge, why not determine extra polygons (or pixels) for independent validation? Or alternatively, if you must split the exiting ESU and expert-identify polygons for training/validation, why not sample by polygons instead of pixels to eliminate the auto-correlation problem. Lastly, I still think a class-based accuracy matrix is needed. An overall accuracy metric is insufficient to suggest good classification from remote sensing images. Reply: This is what we have done. We have now added a class-based accuracy table (Table A1)." No, this Is not what you have done (you randomly sampled 50% pixels, according to the text, meaning pixels from the same polygon are both in training and validation set). Right now, the accuracy and precision are ridiculously high for each class (I would not say this is not possible, but rare) due to the reason you're training and validation all from the same polygon (autocorrelation). What I mean: if you have 216 polygons, you could use 70% of the polygons (all pixels in each polygon should use exclusively for training or validating) for training, and 30% for validation: no training and validation pixels would come from the same polygon. I hope this is clear.

Reply: As discussed above, we have now defined additional and independent polygons, based on expert knowledge. In addition, we found a dataset (Siewert et al. 2016) with independent sampling locations. Those were made for permafrost coring, but included vegetation description that could be linked to our habitat classes. We added text in '3.4.3 Central delta habitat classification' and provide details on accuracy/validation in FigureA4 and Table A1-2, Table S8, Figure S6.

Minor:
Line 275: what is BH?
Reply: We removed the sentence polygons defined by BH.

Figure A4: Adding scales.

Reply: done.

A confusion matrix should include both the user's accuracy and producer's accuracy for each class, as well as the number of samples of each class used for assessment. Refer to Dr. Robert Gilmore Pontius Jr work on "Best Practices for Classification Accuracy Metrics".

Reply: We added the statistical output and class based evaluation in Table A2, using the function confusionMatrix from the R Package caret, an established and frequently used method and output.